# Fusion transcripts FYN-TRAF3IP2 and KHDRBS1-LCK hijack T cell receptor signaling in peripheral T-cell lymphoma, not otherwise specified

Koen Debackere[1,2], Lukas Marcelis[3], Sofie Demeyer[2,4], Marlies Vanden Bempt [1,2,4], Nicole Mentens[2,4], Olga Gielen [2,4], Kris Jacobs[2,4], Michael Broux[2,4], Gregor Verhoef[1,5], Lucienne Michaux[4,6], Carlos Graux[7], Iwona Wlodarska[4,6], Philippe Gaulard[8,9], Laurence de Leval [10], Thomas Tousseyn[3,11], Jan Cools [2,4✉] & Daan Dierickx [1,5✉]

Peripheral T-cell lymphoma (PTCL) is a heterogeneous group of non-Hodgkin lymphomas with poor prognosis. Up to 30% of PTCL lack distinctive features and are classified as PTCL, not otherwise specified (PTCL-NOS). To further improve our understanding of the genetic landscape and biology of PTCL-NOS, we perform RNA-sequencing of 18 cases and validate results in an independent cohort of 37 PTCL cases. We identify *FYN-TRAF3IP2*, *KHDRBS1-LCK* and *SIN3A-FOXO1* as new in-frame fusion transcripts, with *FYN-TRAF3IP2* as a recurrent fusion detected in 8 of 55 cases. Using ex vivo and in vivo experiments, we demonstrate that *FYN-TRAF3IP2* and *KHDRBS1-LCK* activate signaling pathways downstream of the T cell receptor (TCR) complex and confer therapeutic vulnerability to clinically available drugs.

[1] Laboratory for Experimental Hematology, KU Leuven, Leuven, Belgium. [2] Center for Cancer Biology, VIB, Leuven, Belgium. [3] Translational Cell & Tissue Research, KU Leuven, Leuven, Belgium. [4] Center for Human Genetics, KU Leuven, Leuven, Belgium. [5] Department of Hematology, University Hospitals Leuven, Leuven, Belgium. [6] Center for Human Genetics, University Hospitals Leuven, Leuven, Belgium. [7] Mont-Godinne University Hospital, Yvoir, Belgium. [8] Département de Pathologie, Groupe Hospitalier Henri Mondor, AP-HP, Créteil, France. [9] INSERM U955 and Université Paris-Est, Créteil, France. [10] Institute of Pathology, Department of Laboratory Medicine and Pathology, Lausanne University Hospital and Lausanne University, Lausanne, Switzerland. [11] Department of Pathology, University Hospitals Leuven, Leuven, Belgium. ✉email: jan.cools@kuleuven.be; daan.dierickx@uzleuven.be

Peripheral T-cell lymphoma (PTCL) arises from post-thymic mature T lymphocytes. PTCL comprises 10–20% of non-Hodgkin lymphomas, depending on geographical variation. With the exception of *ALK*- and *DUSP22*-translocated anaplastic large cell lymphoma (ALCL)[1], the outcome with current therapies is unsatisfactory[2]. Up to 30% of PTCL cases lack distinctive features and are classified as PTCL, not otherwise specified (PTCL-NOS). Five-year overall survival for PTCL-NOS has stagnated at 30% and guidelines for first-line treatment recommend to treat patients with PTCL-NOS in the context of a clinical trial[2,3]. Although efforts have been made to study the genetic landscape of PTCL-NOS[4–6], the disease biology remains poorly characterized, which hampers the rational design of new clinical trials.

Engagement of immunoreceptors elicits a brisk proliferative and metabolic response in B cells and T cells. This response is intensely intertwined with the Nuclear Factor Kappa light chain enhancer of activated B cells (NF-κB) pathway. Oncogenic activation of the NF-κB pathway by somatic single nucleotide variants and copy number variants is frequent in activated B-cell-like diffuse large B-cell lymphoma[7]. With the notable exception of translocations involving *MALT1* in mucosa-associated lymphoid tissue lymphoma and an *ITK-SYK* gene fusion in peripheral T-cell lymphoma with a T follicular helper phenotype (PTCL-TFH)[8], activation of the NF-κB pathway by gene fusions is rare[7]. On the whole, gene fusions are recurrent disease drivers in hematological malignancies of both lymphoid and myeloid origin. A subset of gene fusions gives rise to chimeric proteins with altered kinase activity or specificity, altered transcriptional regulation or altered post-translational regulation[9,10]. Chimeric transcripts are distinctive disease features with a pivotal role in oncogenesis and have also been identified in PTCL[8,11–14]. These features render them ideal candidates for therapeutic intervention.

In this study, we identify two novel chimeric transcripts in a cohort of PTCL-NOS that hijack signaling pathways downstream of the T cell receptor (TCR) complex. We describe a recurrent *FYN-TRAF3IP2* fusion transcript in PTCL-NOS and PTCL-TFH and a *KHDRBS1-LCK* fusion transcript in PTCL-NOS. Furthermore, we show that both transcripts confer therapeutic vulnerability to clinically available drugs.

## Results

### Transcriptome sequencing identifies fusion transcripts in PTCL-NOS.
To identify fusion events in PTCL-NOS, we performed high coverage, paired-end RNA sequencing (RNA-seq) on a cohort of 15 PTCL-NOS (Supplementary Table 1) cases and 3 PTCL-TFH[13]. Gene expression analysis indicated expression of T follicular helper ($T_{fh}$) markers in the PTCL-TFH cases (Supplementary Fig. 1a). In agreement with the histopathological diagnosis, a subset of PTCL-NOS cases expressed *TNFRSF8* (CD30). Expression of the ALCL marker genes *BATF3* and *TMOD1*[15] clearly separated the PTCL-NOS cases from ALCL cases[12] (Supplementary Fig. 1b).

In 6 cases, we detected fusion transcripts, including the *VAV1-MYO1F* and *TBL1XR1-TP63* fusions (Supplementary Table 1) that were reported previously in PTCL-NOS[11,16], and further confirm the validity of this cohort. In addition, we identified 3 novel in-frame fusion transcripts: *FYN-TRAF3IP2* (2 cases), *KHDRBS1-LCK* (1 case) and *SIN3A-FOXO1* (1 case) (Supplementary Fig. 1c). We found evidence for a gene fusion between the neighboring *FYN* and *TRAF3IP2* genes in 1/15 PTCL-NOS (case PTCL2) and 1/3 PTCL-TFH (case FTCL4). In the PTCL-NOS case, exon 7 of *FYN* was fused to exon 3 of *TRAF3IP2* (Fig. 1a). To exclude that such fusion was the result

of transcriptional read-through, we investigated the genomic region by long-range PCR and we identified an interstitial deletion of 92597 bp on chromosome 6 (Fig. 1b). In the PTCL-TFH case, exon 8 of *FYN* was fused to exon 3 of *TRAF3IP2* (Fig. 1a). In both variants of the fusion protein, the FYN moiety contained the membrane localization motif (SH4 domain) and the SH3 domain of FYN, but lacked the FYN tyrosine kinase domain. The TRAF3IP2 moiety consisted of the almost complete open reading frame, only lacking the first 7 amino acids and thus retaining the TNF receptor-associated factor 6 (TRAF6) binding domain in the fusion protein. Additionally, we discovered a *KHDRBS1-LCK* gene fusion that was caused by an interstitial deletion on chromosome 1 in 1/15 PTCL-NOS (case PTCL17). This gene fusion generated a chimeric transcript in which exon 3 of *KHDRBS1* was fused to exon 3 of *LCK*, resulting in the fusion of the QUA1 and K homology (KH) dimerization domains of KHDRBS1[17] to the SH3, SH2 and kinase domains of LCK (Fig. 1c).

To validate the importance of these fusion genes, we tested the presence of the *FYN-TRAF3IP2* and the *KHDRBS1-LCK* fusion genes with RT-PCR in an independent validation cohort of 37 PTCL cases (30 PTCL-NOS, 6 PTCL-TFH, 1 Adult T-cell leukemia/lymphoma (ATLL)). This screen confirmed that the *FYN-TRAF3IP2* fusion is highly recurrent as it was detected (and verified by Sanger sequencing) in 6/37 (3 PTCL-NOS, 3 PTCL-TFH) samples (Supplementary Fig. 1d, e, Supplementary Table 2). We detected no additional cases of the *KHDRBS1-LCK* fusion in the independent validation cohort.

### FYN-TRAF3IP2-dependent signaling intersects with TCR signaling.
To study the oncogenic properties of the FYN-TRAF3IP2 fusion protein, we cloned the open reading frame (ORF) derived from patient cDNA (case PTCL2) in a bicistronic pMSCV vector with an IRES-GFP reporter (pMIG). As controls, full-length *TRAF3IP2* and the N-terminal fragment of *FYN* containing the membrane localization motif, the SH3 domain and a truncated SH2 domain ($FYN^{1–232}$) were cloned in the pMIG vector (Supplementary Fig. 2a). Only retroviral transduction of interleukin-3 (IL-3) dependent Ba/F3 cells with the *FYN-TRAF3IP2* ORF conferred IL-3 independent growth in vitro, indicating that the FYN-TRAF3IP2 fusion protein is endowed with oncogenic properties and that the combined signaling properties of both fusion partners are required (Fig. 2a).

The oncogenic signaling properties of the FYN-TRAF3IP2 fusion protein could not be related to tyrosine kinase signaling of the FYN kinase domain since it was not retained in the fusion protein. Instead, it was only the N-terminal part of FYN that was fused to the majority of the TRAF3IP2 protein. Under physiological circumstances, TRAF3IP2 is essential for IL-17 signaling. Upon engagement of the heterodimeric IL-17RA/IL-17RC complex, TRAF3IP2 translocates to the juxtamembrane compartment via homotypic SEFIR domain interactions and relays signals to the canonical NF-κB pathway and mitogen-activated protein kinase (MAPK) pathways[18]. *IL-17RA* is ubiquitously expressed, but *IL-17RC* expression is more restricted and defines the cellular response to IL-17[18]. Neither T cells from healthy volunteers (Supplementary Fig. 2b), nor any of the lymphoma samples in our cohort (Supplementary Fig. 2c) expressed *IL17RC*. Accordingly, the levels of TRAF3IP2-regulated transcripts in naive $CD4^+$ T cells transduced with empty pMIG vector (EV), pMIG-*TRAF3IP2* or pMIG-*FYN-TRAF3IP2* were primarily determined by expression of *FYN-TRAF3IP2*. Stimulation with IL-17 had an additive rather than a synergistic effect on the expression TRAF3IP2-regulated transcripts (Fig. 2b). Combined, these data strongly suggest that the

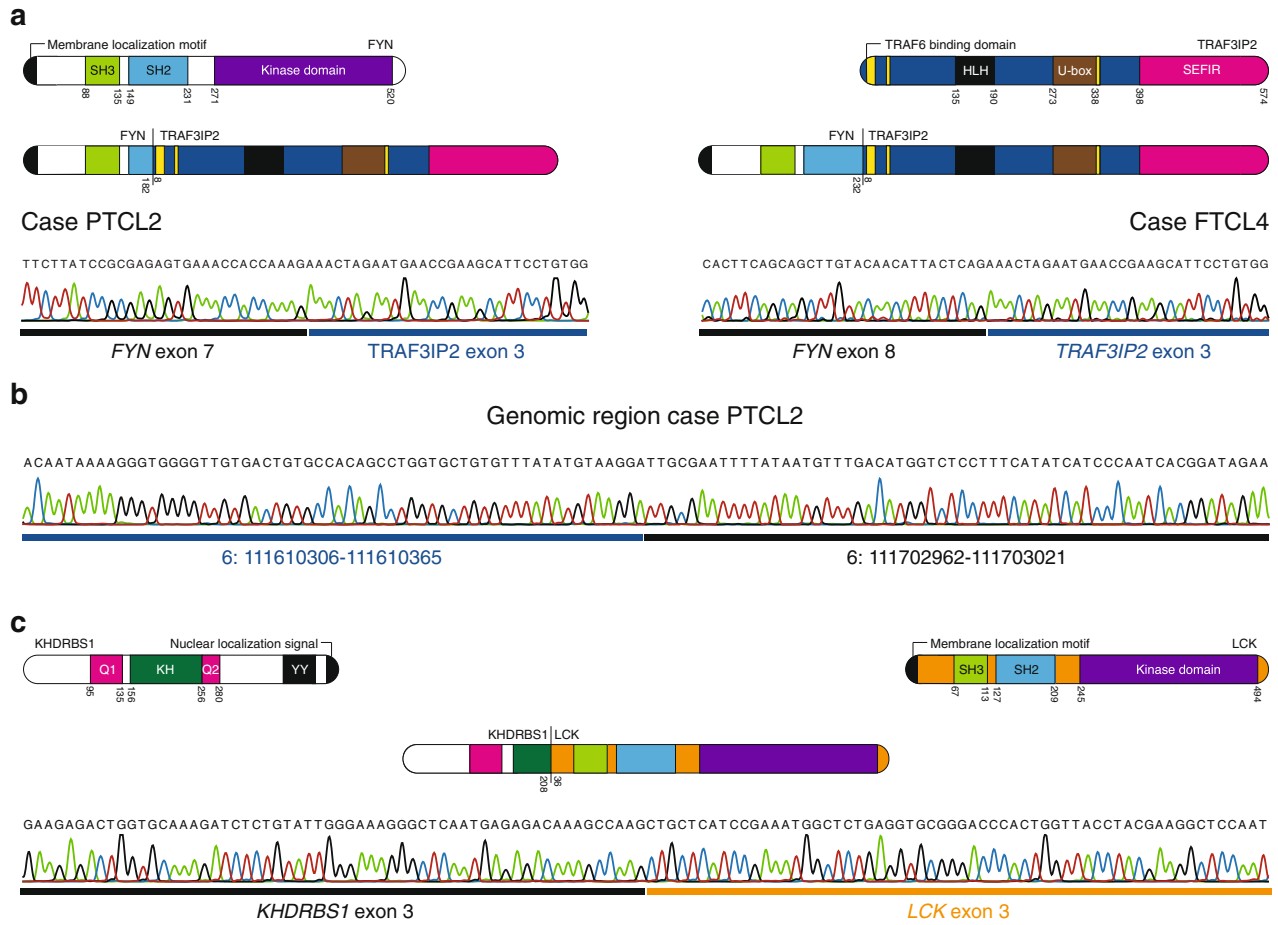

**Fig. 1 FYN-TRAF3IP2 and KHDRBS1-LCK gene fusions in PTCL-NOS. a** Schematic depiction of the protein structure and Sanger sequencing of the RT-PCR amplicon for the *FYN-TRAF3IP2* fusion in the PTCL-NOS case (PTCL2, left) and the PTCL-TFH case (FTCL4, right). **b** Sanger sequencing of the genomic DNA amplicon for the PTCL-NOS case (PTCL2). **c** Schematic depiction of the protein structure and Sanger sequencing of the RT-PCR amplicon for the *KHDRBS1-LCK* fusion.

biological activity of the FYN-TRAF3IP2 fusion protein did not depend on IL-17 signaling.

Given that TCR signaling is a nexus in the integrated control of proliferation and survival in healthy and malignant T cells and TRAF3IP2 activates NF-κB signaling and MAPK downstream of the IL-17 receptor, we hypothesized that aberrant FYN-TRAF3IP2 signaling would intersect with NF-κB signaling and MAPK signaling downstream of the TCR. We transduced the Jurkat T ALL cell line to induce the expression of *TRAF3IP2* or *FYN-TRAF3IP2*. Western blot analysis revealed increased Ser536 phosphorylation of the NF-κB subunit p65 (also known as RelA), indicative of increased activation of the canonical NF-κB pathway. Processing of p100 to p52 and Ser866/870 phosphorylation of p100 did not provide evidence for activation of non-canonical NF-κB signaling (Fig. 2c). Next, we generated a dual GFP/luciferase NF-κB reporter Jurkat cell line to measure NF-κB transcriptional activity. Cells transduced with pMSCV-*FYN-TRAF3IP2*-IRES-*mCherry* (pMImC-*FYN-TRAF3IP2)* had increased NF-κB transcriptional activity in resting conditions and after TCR stimulation with an agonistic CD3 antibody compared to Jurkat cells transduced with pMImC-*FYN*[1–232] or pMImC-*TRAF3IP2* (Fig. 2d). In contrast, we found no evidence for increased activation of MAPK signaling pathways (Fig. 2e).

Finally, we transduced murine CD4[+] T cells to express *TRAF3IP2* or *FYN-TRAF3IP2* (Supplementary Fig. 3a). Intracellular flow cytometry indicated increased activation of canonical NF-κB signaling by means of increased Inhibitor of NF-κB alpha

(IκBα) degradation and increased Ser536 phosphorylation of p65 in *FYN-TRAF3IP2*-expressing cells (Fig. 2f). There was no evidence for enhanced activation of proximal TCR signaling (ZAP70 Tyr318 phosphorylation) or increased MAPK signaling (Fig. 2g).

In summary, these data show that FYN-TRAF3IP2 activates canonical NF-κB signaling independent of IL-17.

**FYN-TRAF3IP2 localizes to the cell membrane and activates the NF-κB pathway.** Ligation of the TCR engenders the formation of a multiprotein signalosome at the membrane-cytoplasm interface. The Src-family kinases FYN and LCK are critical for proximal TCR signaling and are anchored to the plasma membrane by acylation of the unique N-terminal SH4 domains. FYN is myristoylated at Gly2, followed by palmitoylation at Cys3[19]. IL-17 dependent recruitment of TRAF3IP2 to the plasma membrane initiates signaling downstream of TRAF3IP2. Taken together, we reasoned that acylation of FYN-TRAF3IP2 would anchor the fusion protein to the plasma membrane and mediate chronic active TRAF3IP2-dependent signaling.

To this end, we mutated the N-terminal Gly2 of FYN to Ala (FYN[G2A]-TRAF3IP2) to abolish acylation of the N-terminal FYN domain. Immunofluorescence imaging of 293T cells transfected with either wild-type *TRAF3IP2*, *FYN-TRAF3IP2* or *FYN[G2A]-TRAF3IP2*, confirmed that wild-type TRAF3IP2 resided in the cytosol, FYN-TRAF3IP2 segregated to the plasma membrane and FYN[G2A]-TRAF3IP2 resulted in the redistribution of the fusion

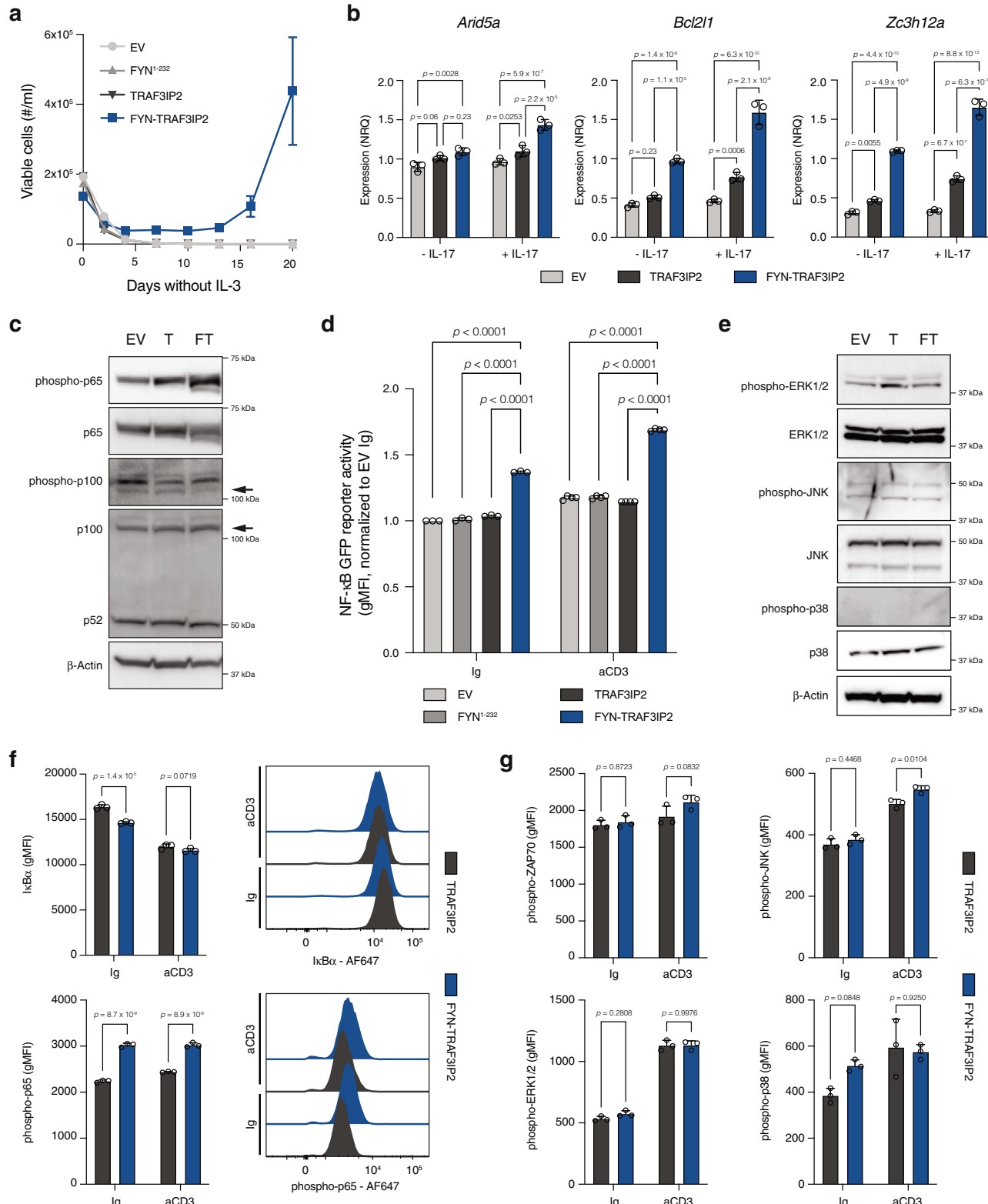

protein to the cytosol (Fig. 3a). The subcellular compartmentalization of FYN-TRAF3IP2 and FYN$^{G2A}$-TRAF3IP2 was confirmed in Ba/F3 cells (Supplementary Fig. 3b) and Ba/F3 cells transduced to express *FYN$^{G2A}$-TRAF3IP2*, were no longer able to grow out in the absence of IL-3 (Fig. 3b).

Next, we collected cytosolic and membrane fractions of Jurkat cells with ectopic expression of *TRAF3IP2*, *FYN-TRAF3IP2* or *FYN$^{G2A}$-TRAF3IP2* and confirmed that disruption of FYN

myristoylation impeded incorporation of the fusion protein in the plasma membrane (Fig. 3c). Delocalization from the plasma membrane impaired the ability of FYN$^{G2A}$-TRAF3IP2 to activate NF-κB transcriptional activity compared to FYN-TRAF3IP2 (Fig. 3d). Likewise, immunofluorescence analysis of CD4$^+$ T cells displayed association of FYN-TRAF3IP2 but not TRAF3IP2 or FYN$^{G2A}$-TRAF3IP2 with the cell membrane (Fig. 3e). Exclusion of FYN-TRAF3IP2 from the membrane

**Fig. 2 FYN-TRAF3IP2-dependent signaling intersects with TCR signaling. a** Outgrowth of Ba/F3 cells transduced with empty pMIG vector, pMIG-*FYN*[1-232], pMIG-*TRAF3IP2* or pMIG-*FYN-TRAF3IP2* after withdrawal of IL-3. $n = 3$ biological replicates per condition. **b** qRT-PCR for IL-17-regulated transcripts in naive CD4[+] T cells transduced with empty pMIG vector, pMIG-*TRAF3IP2* or pMIG-*FYN-TRAF3IP2*. $n = 3$ biological replicates per condition. $p$ values were calculated with Tukey's post-hoc multiple comparisons test. **c** Western blot analysis for canonical and non-canonical NF-κB activation in Jurkat cells transduced with empty pMIG vector, pMIG-*TRAF3IP2* or pMIG-*FYN-TRAF3IP2*. **d** NF-κB GFP reporter signal intensity in Jurkat cells with empty control vector or expression of *FYN*[1-232], *TRAF3IP2* or *FYN-TRAF3IP2* after stimulation with an agonistic anti-CD3ε antibody or exposure to control immunoglobulin (Ig). $n = 3$ biological replicates per condition for Ig control, $n = 4$ for anti-CD3ε. $p$ values were calculated with Tukey's post-hoc multiple comparisons test. **e** Western blot analysis for activation of MAPK pathways in Jurkat cells transduced with empty pMIG vector, pMIG-*TRAF3IP2* or pMIG-*FYN-TRAF3IP2*. **f** Intracellular flow cytometry for IκBa (top row) or phospho-p65 (bottom row) in CD4[+] T cells treated with control immunoglobulin (Ig) or agonistic anti-CD3e antibody. $n = 3$ biological replicates per condition. $p$ values were calculated with Šidák's multiple comparisons test (two-sided). **g** Intracellular flow cytometry proximal TCR signaling and MAPK pathway activation in CD4[+] T cells treated with control immunoglobulin (Ig) or agonistic anti-CD3e antibody. $n = 3$ biological replicates per condition. $p$ values were calculated with Šidák's multiple comparisons test (two-sided). All data are represented as mean ± standard deviation (SD).

compartment, impaired activation of NF-κB signaling in CD4[+] T cells (Fig. 3f).

Together, these experiments demonstrate that acylation of the N-terminal SH4 domain of FYN anchors the FYN-TRAF3IP2 fusion protein to the plasma membrane. Partitioning of FYN-TRAF3IP2 to the plasma membrane is required for downstream activation of NF-κB signaling.

**FYN-TRAF3IP2 activates TRAF6 independent of the CBM signalosome.** Activation of the NF-κB pathway downstream of the TCR is initiated by protein kinase C θ (PKCθ)-dependent assembly of a signalosome composed of CARD11, BCL10 and MALT1 (CBM signalosome). The CBM signalosome recruits TRAF6, which leads to lysine 63-linked (K63-linked) poly-ubiquitination of TRAF6. K63-linked polyubiquitinated TRAF6 acts as a scaffold for the IκB Kinase (IKK) complex and Transforming-growth-factor-β-Activated Kinase 1 (TAK1) complex which will ultimately lead to the release of NF-κB transcription factors from IκB[20]. The TRAF3IP2 protein comprises an N-terminal TRAF6 binding motif[21] and has a U-box domain with E3 ubiquitin ligase enzymatic activity through which it catalyzes K63-linked polyubiquitination of TRAF6[22] (Fig. 1a). We assumed that the interaction of the FYN-TRAF3IP2 fusion protein with TRAF6 would be preserved and enable TRAF6 K63-linked polyubiquitination independent of the CBM signalosome to activate NF-κB.

To test this hypothesis, we substituted the acidic amino acid residues in the N-terminal TRAF6 binding motif (PVEVDE) with Ala residues (PVAVAA). While wild-type TRAF3IP2 and FYN-TRAF3IP2 co-immunoprecipitated with TRAF6, mutation of the TRAF6 binding motif (FYN-TRAF3IP2[ΔT6]) abrogated the interaction with TRAF6 and led to a decrease in K63-linked polyubiquitination of TRAF6 (Fig. 4a). In the opposite direction, co-immunoprecipitation of TRAF6 – including polyubiquitinated TRAF6 – with FYN-TRAF3IP2[ΔT6] was attenuated (Fig. 4b). Contrary to Ba/F3 cells with expression of *FYN-TRAF3IP2*, Ba/F3 cells with ectopic expression of *FYN-TRAF3IP2[ΔT6]* (Supplementary Fig. 3c) did not grow out after withdrawal of IL-3 (Fig. 4c). Expression of *FYN-TRAF3IP2[ΔT6]* in Jurkat cells or primary CD4[+] T cells (Supplementary Fig. 3c) impaired activation of the canonical NF-κB pathway (Fig. 4d, e).

To investigate whether FYN-TRAF3IP2 could activate the NF-κB pathway independent of CARD11, we generated *CARD11* knock-out Jurkat cells with CRISPR/Cas9 genome editing (Supplementary Fig. 3d, e). Expression of *FYN-TRAF3IP2*, but not *TRAF3IP2* or *FYN-TRAF3IP2[ΔT6]* augmented NF-κB transcriptional activity in resting conditions in both wild-type and *CARD11* knock-out Jurkat cells. Activation of PKCθ with phorbol 12-myristate 13-acetate (PMA) and ionomycin increased NF-κB transcriptional activity in wild-type Jurkat cells, but significantly

more in Jurkat cells with expression of *FYN-TRAF3IP2*. CARD11 deficiency impaired NF-κB activation in response to PMA and ionomycin, but *CARD11* knock-out Jurkat cells with *FYN-TRAF3IP2* expression remained significantly more responsive to PMA/ionomycin stimulation than empty vector control or cells with expression of *TRAF3IP2* or *FYN-TRAF3IP2[ΔT6]*. This proves that FYN-TRAF3IP2 does not require the CBM signalosome, but suggests an interaction – direct or indirect – between FYN-TRAF3IP2 and PKCθ.

Collectively, these results indicate that FYN-TRAF3IP2 directly interacts with TRAF6 and activates TRAF6 and the NF-κB pathway without intervention of the CBM signalosome.

**FYN-TRAF3IP2 expression causes PTCL-NOS-like disease in vivo.** To test the oncogenic potential of the fusion protein in vivo, we transduced lineage negative hematopoietic stem and progenitor cells (HSPC) from wild-type mice with either the empty pMIG vector or the pMIG vector containing *FYN-TRAF3IP2* or *FYN[G2A]-TRAF3IP2*. Subsequently, we injected the transduced cells (Supplementary Fig. 4a) intravenously in sub-lethally irradiated syngeneic recipient mice. Engraftment rates were comparable (Supplementary Fig. 4b). As early as 7 weeks after transplantation, GFP[+] cells in the peripheral blood of mice transplanted with HSPC expressing *FYN-TRAF3IP2* skewed towards CD4[+] cells (Supplementary Fig. 4c). Mice with expression of *FYN-TRAF3IP2* in hematopoietic cells, started losing weight 8 weeks after transplantation and succumbed within 16 weeks after transplantation (Fig. 5a). These mice had spleno-megaly (Fig. 5b) and generalized lymphadenopathy. In contrast, mice transplanted with cells expressing mutant *FYN[G2A]-TRAF3IP2* did not develop disease (Fig. 5a, b). Lymph nodes from mice with *FYN-TRAF3IP2*-expressing cells displayed an effaced lymph node architecture with paracortical expansion and loss of germinal centers. The lymph nodes contained medium-sized and large cells with irregular, pleomorphic nuclei and multiple, pro-minent nucleoli. Malignant cells were surrounded by a poly-morphic infiltrate with numerous eosinophils and histiocytes (Fig. 5c). *FYN-TRAF3IP2*-driven lymphomas were invariably CD4[+] (Fig. 5d). These cells were also found in the spleens (Supplementary Fig. 4d) and infiltrated the livers (Fig. 5e, f). FYN-TRAF3IP2 and FYN[G2A]-TRAF3IP2 were expressed in lymph nodes (Fig. 5g) and FYN-TRAF3IP2 expression was restricted to GFP[+] cells (Supplementary Fig. 4e). Analysis of *Tcrb* locus rearrangements with RT-PCR corroborated the clonal nature of the disease (Fig. 5h).

Because we identified *FYN-TRAF3IP2* gene fusions in both PTCL-NOS and PTCL-TFH, we determined the immunophenotype of murine *FYN-TRAF3IP2*-driven lymphomas. Malignant cells were positive for PD-1 and ICOS, but negative for the T_fh marker CXCR5 (Fig. 6a). The BCL6 transcription factor orchestrates T_fh

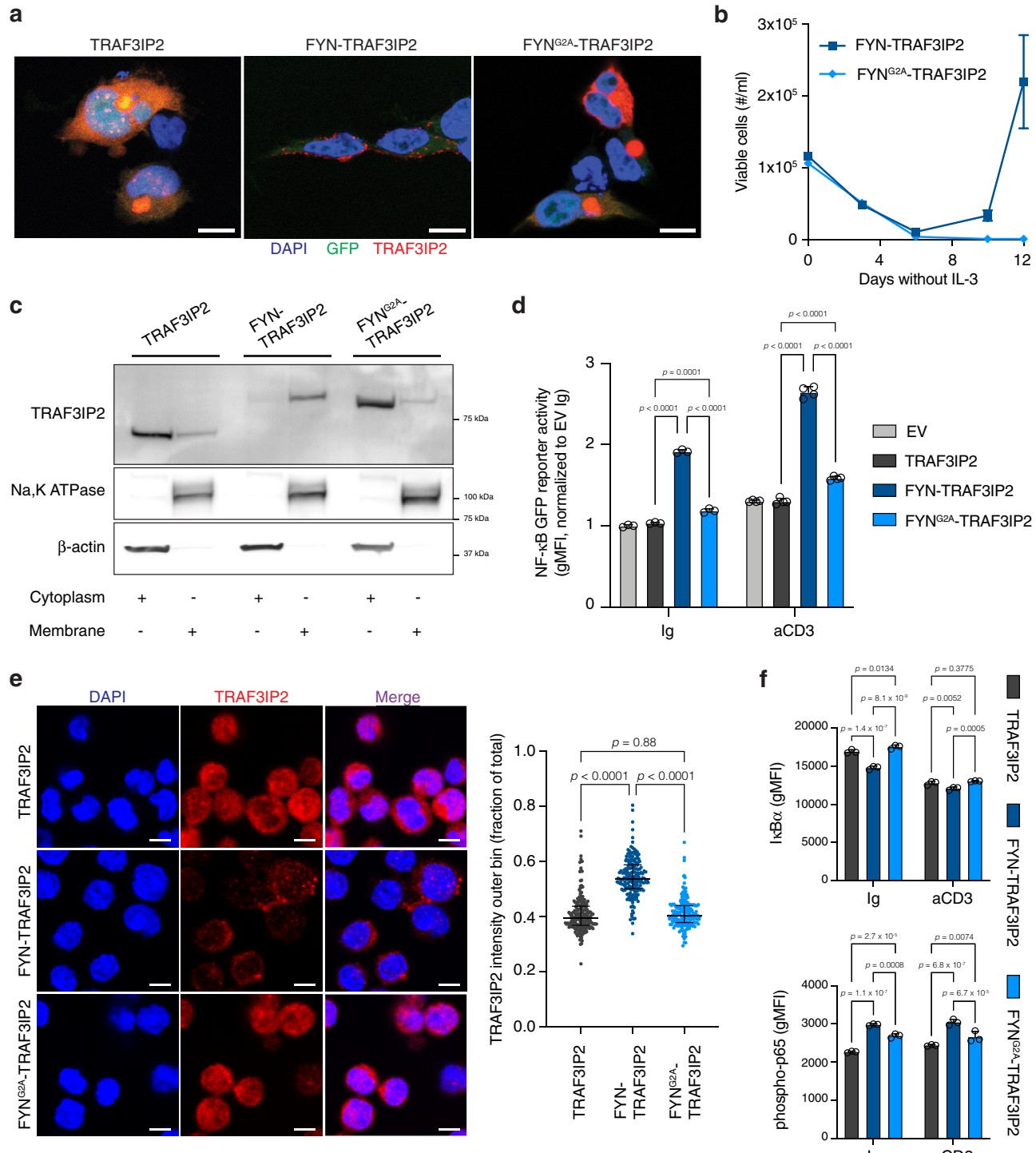

**Fig. 3 Membrane anchoring of FYN-TRAF3IP2 is required to activate NF-κB signaling. a** Immunofluorescence images of 293T cells transfected with *TRAF3IP2*, *FYN-TRAF3IP2* and *FYN^G2A^-TRAF3IP2* stained for TRAF3IP2. Scalebars represent 10 μm. **b** Outgrowth of Ba/F3 cells transduced with pMIG-*FYN-TRAF3IP2* or pMIG-*FYN^G2A^-TRAF3IP2* after withdrawal of IL-3. $n = 3$ biological replicates per condition. **c** Western blot analysis for TRAF3IP2 on cytoplasmic and membrane fractions of Jurkat cells transduced with pMIG-*TRAF3IP2*, pMIG-*FYN-TRAF3IP2* and pMIG-*FYN^G2A^-TRAF3IP2*. **d** NF-κB GFP reporter signal intensity in Jurkat cells with empty control vector or expression of TRAF3IP2, FYN-TRAF3IP2 or FYN^G2A^-TRAF3IP2 after stimulation with an agonistic anti-CD3ε antibody or exposure to control immunoglobulin (Ig). $n = 3$ biological replicates per condition for Ig control, $n = 4$ for anti-CD3ε. *p* values were calculated with Tukey's post-hoc multiple comparisons test. **e** Immunofluorescence images (left) of TRAF3IP2 staining on CD4^+^ T cells transduced to express TRAF3IP2, FYN-TRAF3IP2 or FYN^G2A^-TRAF3IP2. Scalebars represent 5 μm. Quantification (right) of TRAF3IP2 signal intensity in the outer cell bin, expressed as fraction of signal intensity over the entire cell. Each dot represents a cell ($n = 201$ for TRAF3IP2, $n = 192$ for FYN-TRAF3IP2 and $n = 179$ for FYN^G2A^-TRAF3IP2). Horizontal line and whiskers represent median and interquartile range respectively. *p* values were calculated with Games-Howell's multiple comparisons test. **f** Intracellular flow cytometry for IκBa (top row) or phospho-p65 (bottom row) in CD4^+^ T cells treated with control immunoglobulin (Ig) or agonistic anti-CD3ε antibody. $n = 3$ biological replicates per condition. *p* values were calculated with Tukey's post-hoc multiple comparisons test. All data are represented as mean ± SD unless stated otherwise.

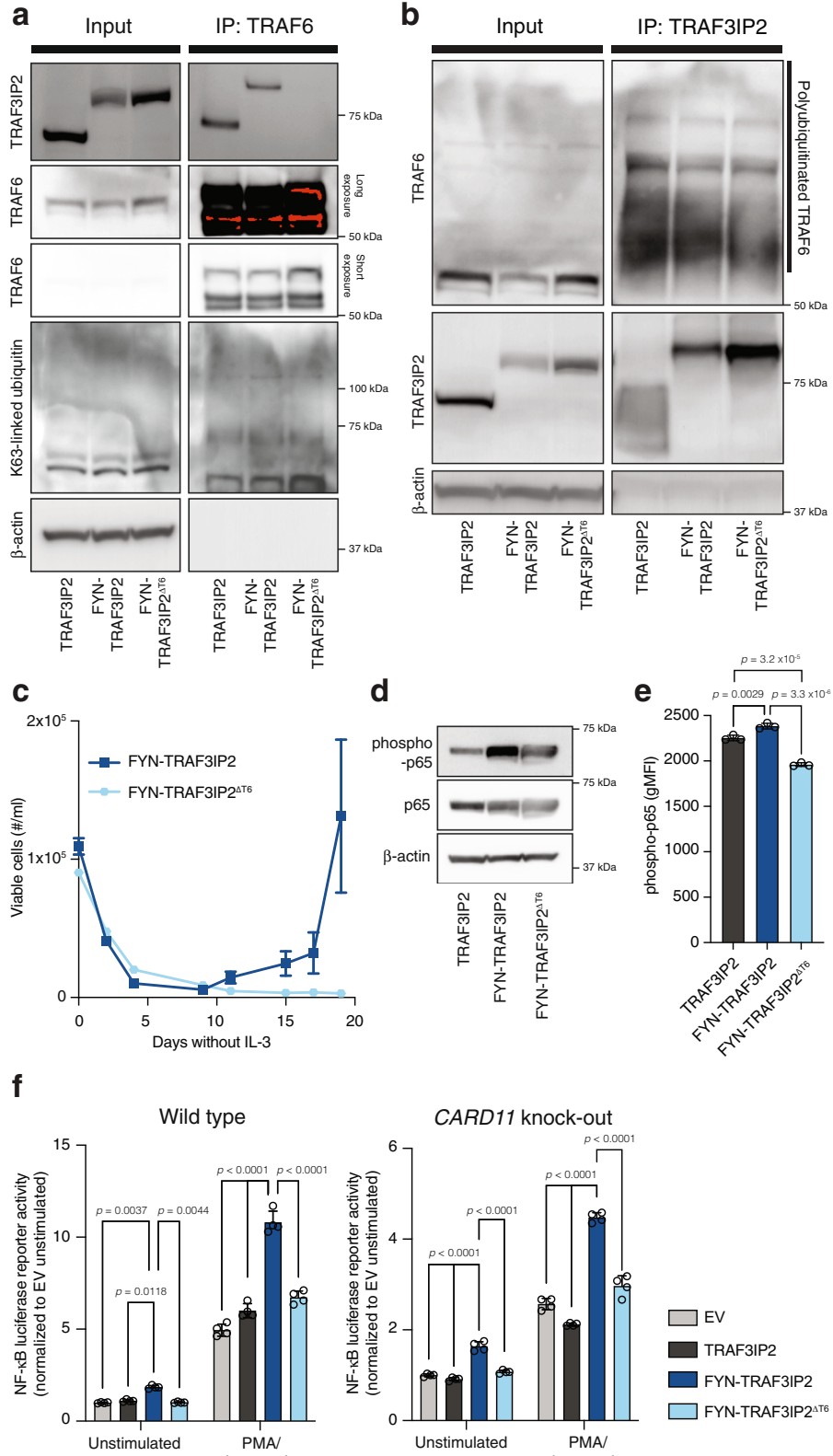

identity[23], but was not expressed in *FYN-TRAF3IP2*-induced lymphomas (Fig. 6b). B cell proliferation with an expansion of germinal center (GC) B cells and plasma cells frequently accompanies T_{fh}-related neoplasms. In contrast, we observed a reduction in B cells in *FYN-TRAF3IP2*-associated lymphomas (Fig. 6c) without an increased proportion of GC B cells (Fig. 6d). The presence of plasma cells was more variable, but overall, not

significantly altered (Fig. 6e). Consistent with the histologic finding of an increased number of histiocytes and eosinophils, the proportion of CD11b[+] myeloid cells was increased in *FYN-TRAF3IP2*-driven lymphomas (Supplementary Fig. 4f). The fraction of CD4[+] cells was increased in *FYN-TRAF3IP2*-driven lymphomas and there were no changes in the number of CD8[+] cells (Supplementary Fig. 4f). Finally, there was no proliferation of high

**Fig. 4 FYN-TRAF3IP2 activates TRAF6 and canonical NF-κB signaling independent of the CBM signalosome. a** Western blot analysis of whole-cell lysate and anti-TRAF6 immunoprecipitate in TRAF3IP2-, FYN-TRAF3IP2- and FYN-TRAF3IP2$^{\Delta T6}$-expressing Jurkat cells. **b** Western blot analysis of whole-cell lysate and anti-TRAF3IP2 immunoprecipitate in TRAF3IP2-, FYN-TRAF3IP2- and FYN-TRAF3IP2$^{\Delta T6}$-expressing Jurkat cells. **c** Outgrowth of Ba/F3 cells transduced with pMIG-FYN-TRAF3IP2 or pMIG-FYN-TRAF3IP2$^{\Delta T6}$ after withdrawal of IL-3. $n = 3$ biological replicates per condition. **d** Western blot analysis for canonical NF-κB activation in Jurkat cells transduced with pMIG-TRAF3IP2, pMIG-FYN-TRAF3IP2 or pMIG-FYN-TRAF3IP2$^{\Delta T6}$. **e** Intracellular flow cytometry for phospho-p65 in resting CD4$^+$ T cells with expression of TRAF3IP2, FYN-TRAF3IP2 or FYN-TRAF3IP2$^{\Delta T6}$ . $n = 3$ biological replicates per condition. $p$ values were calculated with Tukey's post-hoc multiple comparisons test. **f** NF-κB luciferase reporter signal intensity in wild-type Jurkat cells (left) or CARD11 knock-out Jurkat cells (right) with empty control vector or expression of TRAF3IP2, FYN-TRAF3IP2 or FYN-TRAF3IP2$^{\Delta T6}$ in resting conditions or after stimulation with PMA. $n = 4$ biological replicates per condition. $p$ values were calculated with Tukey's post-hoc multiple comparisons test. All data are represented as mean ± SD.

endothelial venules (Fig. 6f). These results are compatible with PTCL-NOS rather than PTCL-TFH.

Collectively, these in vivo data indicate that *FYN-TRAF3IP2* is a disease driver in PTCL-NOS and that the oncogenic potential of the FYN-TRAF3IP2 fusion protein is critically dependent on its association with the plasma membrane because lymphomagenesis was abolished in *FYN$^{G2A}$-TRAF3IP2*-expressing cells.

**FYN-TRAF3IP2-driven lymphomas have active NF-κB signaling and are sensitive to inhibition of BCL-X$_L$.** We compared the gene expression profile of sorted CD4$^+$GFP$^+$ lymphoma cells with naive CD4$^+$ T cells, CD4$^+$GFP$^-$ non-malignant stromal T cells from the lymphoma and CD4$^+$GFP$^+$ *FYN$^{G2A}$-TRAF3IP2*-expressing T cells. Consistent with the neoplastic nature of the disease, numerous cell-cycle-associated genes were among the most significantly upregulated genes in *FYN-TRAF3IP2*-expressing lymphoma cells (Supplementary Fig. 5a). Overexpression of *Icos* and *Pdcd1* was congruent with the immunophenotype determined with flow cytometry. Additionally, lymphoma cells overexpressed *Runx2* and *Id2*. Both genes oppose the T$_{fh}$ phenotype and are repressed by *Bcl6*[23], in support of a non-T$_{fh}$ origin of the lymphomas (Supplementary Fig. 5a). We analyzed the cis-regulatory features associated with differentially expressed genes to identify the transcription factors driving these phenotypes with i-cisTarget. This analysis consistently retrieved the NF-κB1 motif and the PU.1 motif as the most significantly enriched cis-regulatory features associated with overexpressed genes in *FYN-TRAF3IP2*-expressing lymphoma cells (Fig. 7a). Gene set enrichment analysis (GSEA) for different p65 (RelA) target gene sets reaffirmed increased expression of canonical NF-κB target genes in *FYN-TRAF3IP2*-expressing cells (Fig. 7b) and nuclear translocation of p65 was most pronounced in *FYN-TRAF3IP2*-expressing cells (Fig. 7c). As expected, malignant cells expressed *Relb* – a known direct transcriptional target of p50/p65[24] (Supplementary Fig. 5b). However, there was no enrichment of RelB target genes in malignant cells (Supplementary Fig. 5c) and RelB was sequestered in the cytosol of *FYN-TRAF3IP2*-expressing cells (Supplementary Fig. 5b). The transcription factor PU.1 (also known as SPI1) has been associated with the T$_h$9 phenotype. The cytokine profile of *FYN-TRAF3IP2*-expressing lymphoma cells was compatible with a T$_h$9 phenotype[25] (Supplementary Fig. 5d). Consistent with in vitro assays, there was no univocal activation of MAPK signaling in *FYN-TRAF3IP2*-expressing cells. The AP-1 motif was associated with cis-regulatory features of upregulated genes in lymphoma cells versus naive CD4$^+$ T cells and downregulated genes in lymphoma cells versus *FYN$^{G2A}$-TRAF3IP2*-expressing CD4$^+$ T cells (Fig. 7a). Likewise, GSEA for c-Jun target genes yielded inconsistent results (Supplementary Fig. 5e).

Targeting NF-κB directly for clinical purposes remains elusive because of on-target toxicities. This prompted us to examine vulnerabilities downstream of NF-κB. Promoting cell survival through induction of target genes, is one of the best-documented functions of NF-κB[26]. Indeed, pro-survival and to a lesser extent pro-apoptotic factors were upregulated in *FYN-TRAF3IP2*-expressing cells (Fig. 8a). We reasoned that inhibition of pro-survival signals could tilt the balance towards apoptosis. BCL-X$_L$ (encoded by the *Bcl2l1* gene) was consistently overexpressed by lymphoma cells at both the RNA level and protein level in our mouse model (Fig. 8a, b). BCL-X$_L$ expression was also confirmed in case PTCL2 (Fig. 8c). Compared to naive CD4$^+$ T cells cultured under identical conditions, *FYN-TRAF3IP2*-expressing cells were >30-fold more sensitive to inhibition of BCL-X$_L$, BCL-W and BCL2 with ABT-263 treatment (Fig. 8d).

These results demonstrate that *FYN-TRAF3IP2* activates NF-κB signaling in vivo and that this confers malignant cells with vulnerability to inhibition of BCL-X$_L$.

**KHDRBS1-LCK mediates chronic active TCR signaling.** We cloned the open reading frame of the *KHDRBS1-LCK* fusion transcript derived from patient cDNA in the pMIG vector to study the signaling properties of the KHDRBS1-LCK fusion protein. Here, the kinase domain of LCK was fused to the KHDRBS1 dimerization domain. It thus lacks the nuclear localization signal of KHDRBS1 and the membrane localization motif of LCK (Fig. 1c), suggesting that this protein functions in the cytosol. Indeed, in primary T cells with ectopic expression of *KHDRBS1-LCK*, the fusion protein accumulated in the cytosol when compared to T cells with ectopic expression of *LCK* (Fig. 9a). Retroviral transduction of IL-3 dependent Ba/F3 cells with *KHDRBS1-LCK* (Supplementary Fig. 6a) led to IL-3 independent growth in vitro and this was abolished by an inactivating LCK kinase domain mutation (*KHDRBS1-LCK$^{K273R}$*)[27] (Fig. 9b). Likewise, inhibition of LCK kinase activity with dasatinib could efficiently block the proliferation of transformed Ba/F3 cells at low nanomolar concentrations (Fig. 9c). KHDRBS1-LCK but not KHDRBS1-LCK$^{K273R}$ was constitutively active (Tyr394 phosphorylation) in unstimulated Jurkat cells (Fig. 9d). Likewise, proximal TCR signaling was significantly enhanced in primary T cells with expression of *KHDRSB1-LCK* in resting conditions and after TCR stimulation compared to primary T cells transduced with empty pMIG vector or *LCK* (Fig. 9e). Inhibition of LCK kinase activity with dasatinib reverted these differences (Fig. 9e). Further downstream of the TCR, KHDRBS1-LCK activated MAPK signaling and this was inhibited by dasatinib (Fig. 9f).

Collectively, these data show that KHDRBS1-LCK is a constitutively active tyrosine kinase that leads to chronic active TCR signaling.

**KHDRBS1-LCK instigates PTCL-NOS in vivo.** To test the oncogenic potential of *KHDRBS1-LCK* in vivo, we transduced HSPC from wild-type mice with MSCV retrovirus with either an empty pMIG vector or a pMIG vector containing the

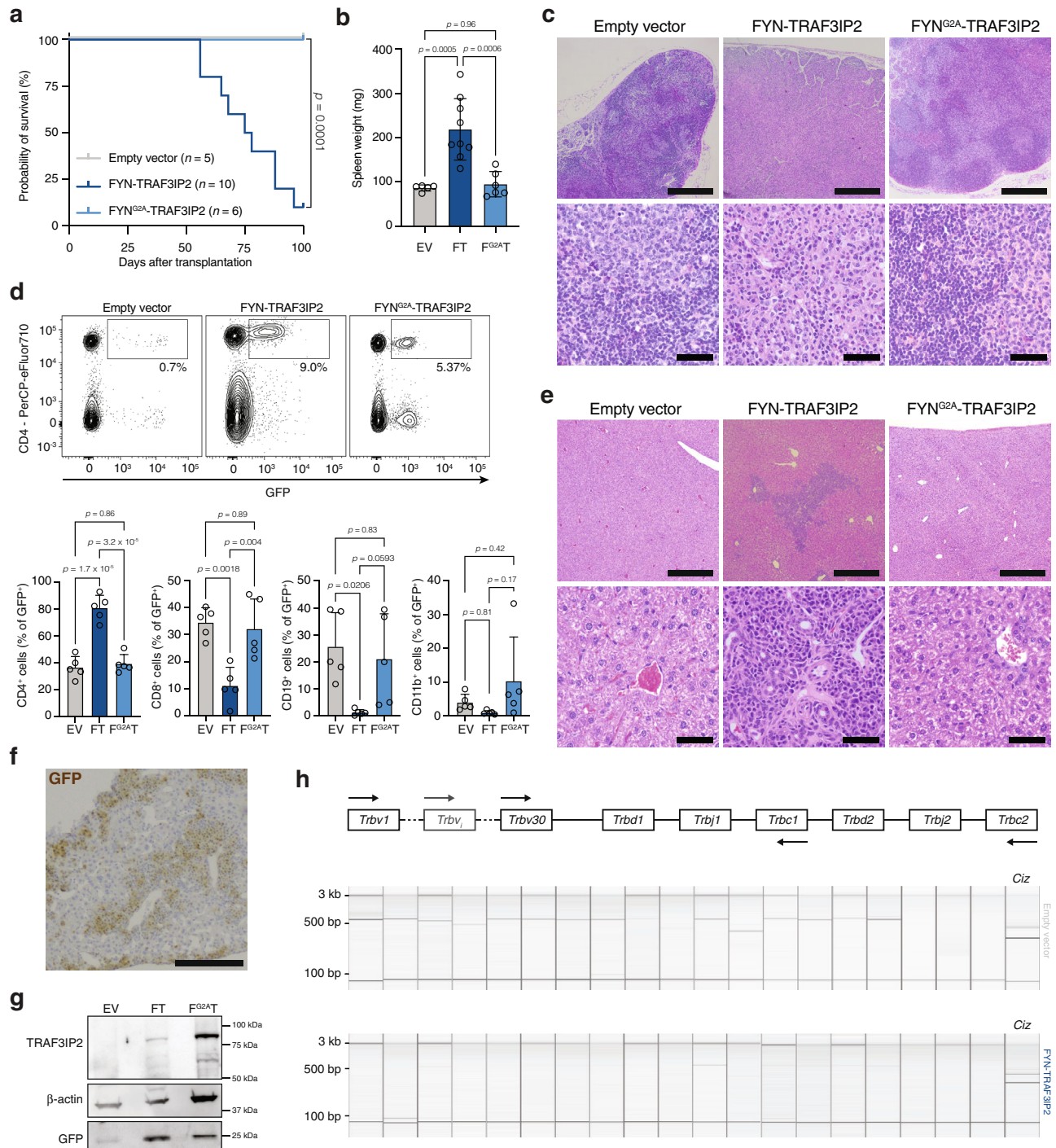

KHDRBS1-LCK ORF. We observed incomplete disease penetrance

KHDRBS1-LCK ORF. We observed incomplete disease penetrance (Fig. 10a). One mouse developed a lymphoproliferative disorder with generalized lymphadenopathy, moderate splenomegaly, pleural effusion and thymic enlargement, but no leukocytosis (Supplementary Fig. 6b). Normal tissue architecture was effaced. Residual follicles were pushed to the border of the lymph nodes. The expanded paracortex was dominated by a monotonous proliferation of medium-sized lymphocytes with irregular nuclei with 1 or 2 nucleoli and an increased number of blood vessels (Fig. 10b). Malignant cells were CD4−CD8+ and the majority expressed surface TCRβ (Fig. 10c, Supplementary Fig. 6c). Bone marrow infiltration was limited. Negativity for TdT ruled out a lymphoblastic lymphoma/leukemia (Fig. 10d). Lymphomas expressed the

KHDRBS1-LCK fusion protein, which was highly phosphorylated (Fig. 10e). Interestingly, also endogenous LCK was highly phosphorylated in lymphomatous spleens compared to spleens from empty pMIG vector mice despite comparable LCK protein levels, hinting at activation of endogenous LCK by KHDRBS1-LCK.

## Discussion

Recent advances on the genetics of PTCL-NOS have not led to tangible results for the development of novel therapeutic strategies. On the other hand, a handful of therapies (e.g., romidepsin[28]) is successful in a minority of patients with PTCL-NOS, but insight in their mechanism of activity is lacking. In this

**Fig. 5 Expression of *FYN-TRAF3IP2* in hematopoietic cells fosters the development of PTCL-NOS. a** Kaplan–Meier survival curve after transplantation with HSPC transduced with empty pMIG vector ($n = 5$), pMIG-*FYN-TRAF3IP2* ($n = 10$) or pMIG-*FYN^{G2A}-TRAF3IP2-GFP* ($n = 6$). Log-rank $p$ value was obtained from a two-sided Chi-square test. **b** Spleen weights at sacrifice of mice transplanted with HSPC transduced with empty vector (EV, $n = 5$), *FYN-TRAF3IP2* (FT, $n = 9$) or *FYN^{G2A}-TRAF3IP2* (F^{G2A}T, $n = 6$). $p$ values were calculated with Tukey's post-hoc multiple comparisons test. **c** Representative H&E stains of lymph nodes from mice transplanted with HSPC transduced with empty vector ($n = 5$), *FYN-TRAF3IP2* ($n = 5$) or *FYN^{G2A}-TRAF3IP2* ($n = 5$). Low magnification images are in the top row, scalebars represent 500 μm. High magnification images are in the bottom row, scalebars represent 50 μm. **d** Representative flow cytometry plots for cell suspensions from the lymph nodes of EV mice, FT mice and F^{G2A}T mice (top). Quantification of the lineage of GFP^+ cells in lymph node cell suspensions (bottom). $n = 5$ mice per group. $p$ values were calculated with Tukey's post-hoc multiple comparisons test. **e** Representative H&E stains of livers from mice transplanted with HSPC transduced with empty vector ($n = 5$), *FYN-TRAF3IP2* ($n = 5$) or *FYN^{G2A}-TRAF3IP2* ($n = 5$). Low magnification images are in the top row, scalebars represent 500 μm. High magnification images are in the bottom row, scalebars represent 50 μm. **f** Representative picture of GFP immunohistochemistry on liver sections of mice with *FYN-TRAF3IP2*-expressing cells ($n = 5$). Scalebar represents 200 μm. **g** Western blot analysis for TRAF3IP2 and GFP on lymph node lysates after transplantation with HSPC transduced with empty vector, pMIG-*FYN-TRAF3IP2* or pMIG-*FYN^{G2A}-TRAF3IP2*. **h** Representative gel analysis pictures for RT-PCR analysis of *Tcrb* rearrangements (left to right: *Trbv6, Trbv1, Trbv26, Trbv2, Trbv12-1/12-2/12-3, Trbv19, Trbv29, Trbv13-1/13-2/13-3, Trbv17, Trbv4, Trbv16, Trbv15, Trbv14, Trbv31, Trbv20, Trbv3, Trbv18, Trbv30* and *Trbv21*) and *Ciz* control reaction (far right lane) on lymph nodes from an empty vector control mouse (top, $n = 2$) and a FYN-TRAF3IP2-induced lymphoma (bottom, $n = 5$). All data are represented as mean ± SD.

study, we have identified two novel gene fusions in PTCL-NOS that could function as targets for therapy. We have studied their activity in T-cell models, both in vitro and in vivo, and we were able to show that both gene fusions confer therapeutic vulnerability to clinically available compounds.

We identify and characterize here a recurrent *FYN-TRAF3IP2* gene fusion in PTCL-NOS and PTCL-TFH. Despite the apparent heterogeneity of PTCL-NOS, the *FYN-TRAF3IP2* gene fusion ranks as one of the most recurrent events identified to date in this disease. To gain insight in the biology of malignant cells, efforts have been made to establish a cell of origin for different T-cell lymphomas. PTCL-NOS has been divided in subgroups based on the expression of the T$_h$2-related transcription factor *GATA3* or the T$_h$1- and cytotoxic-T-cell-related transcription factors *TBX21* and *EOMES*[4]. In our mouse model there was evidence for a T$_h$9-related signature in *FYN-TRAF3IP2*-driven lymphomas. Except for one CD8^+ and one CD4/CD8 double-negative PTCL-NOS, all human samples with a *FYN-TRAF3IP2* gene fusion were CD4^+.

FYN-TRAF3IP2 activates canonical NF-κB signaling to exert its oncogenic effects. Activation of NF-κB by both cell-intrinsic events and cell-extrinsic cues is well described for various B-cell lymphomas and multiple myeloma[7]. On the contrary, the contribution of NF-κB signaling to T-cell lymphomagenesis remains ill-defined. The relative contribution of canonical and non-canonical NF-κB signaling to PTCL-NOS biology remains largely unresolved and studies reached conflicting conclusions[29,30]. Mechanistic studies provided evidence for both pathways in molecularly defined subsets of PTCL. In ATLL, *PRKCB* mutations activate canonical NF-κB[31] and IRF4 and p65 jointly dictate super-enhancer formation[32]. On the other hand, non-canonical NF-κB signaling is activated downstream of CD30 in CD30^+ PTCL[33]. In ALCL, RelB enhances transcriptional activation by *NFKB2-ROS1* gene fusions[12] and STAT3 drives the expression of *CD30* and *NFKB2*[34]. Overexpression of *GAPDH* in T cells, leads to a PTCL-TFH-like disease in mice which is largely driven by non-canonical NF-κB signaling[35]. We show that the oncogenic properties of FYN-TRAF3IP2 are primarily determined by its ability to activate canonical NF-κB without a significant contribution of non-canonical NF-κB.

In physiological circumstances, TRAF3IP2-dependent signaling is initiated after recruitment to the plasma membrane upon IL-17 receptor engagement[18]. We show that the FYN-TRAF3IP2 fusion protein undergoes acylation at the FYN-derived N-terminal SH4-domain. This post-translational modification anchors the protein to the plasma membrane and renders it constitutively active. We identify the canonical NF-κB pathway as the prime target of FYN-TRAF3IP2. Likewise, the pleckstrin homology

domain of ITK directs the ITK-SYK fusion protein to the plasma membrane leading to constitutive SYK kinase activity and chronic active TCR signaling in PTCL-TFH[8]. Unlike ITK-SYK, FYN-TRAF3IP2 does not have kinase activity, but acts as a signaling adapter and has E3 ubiquitin ligase activity[22]. Together with the Ubc13-Uev1A E2 complex[22], FYN-TRAF3IP2 will catalyze K63-linked polyubiquitination of TRAF6. K63-linked polyubiquitinated TRAF6 will activate the IKK and TAK1 kinase complexes. Phosphorylation by these complexes, will mark IκB for proteasomal degradation thereby releasing NF-κB transcription factors. These are now free to translocate to the nucleus and initiate transcription. Importantly, our data suggest that the direct interaction between FYN-TRAF3IP2 and TRAF6 will bypass most of the paradigmatic TCR cascade, including the CBM signalosome.

Several outstanding questions remain to be answered. First, the *FYN-TRAF3IP2* gene fusion appears specific for mature T cell neoplasms. It was not identified in recent large-scale studies of mature B cell neoplasms[36–38]. A concurrent report confirms that the *FYN-TRAF3IP2* gene fusion is a T-cell lymphoma exclusive event[39]. In our mouse model all mice invariably developed CD4^+ T-cell lymphomas despite constitutive expression of the fusion protein in all hematopoietic lineages. T cell specificity could be related to interactions of the FYN-TRAF3IP2 protein with T cell-specific partners. Indeed, the SH4 domain of FYN can interact directly with CD3ζ[40]. Therefore, it is plausible that the FYN-TRAF3IP2 fusion protein preferentially partitions to TCR cluster regions in the plasma membrane, facilitating the activation of downstream pathways through vicinity. Our data suggest that FYN-TRAF3IP2-initiated signaling bypasses proximal TCR components such as ZAP70. Further downstream in the TCR cascade, *FYN-TRAF3IP2*-expressing cells display enhanced responsiveness to activation of PKCθ by PMA/ionomycin, but do not require the CBM signalosome to activate TRAF6. Rather FYN-TRAF3IP2 interacts directly with TRAF6 to activate canonical NF-κB signaling. From these observations, it is reasonable to assume that FYN-TRAF3IP2 requires proximity to the TCR complex to activate NF-κB. FYN-TRAF3IP2-dependent signaling appears to bypass much of the TCR cascade, but our findings could point at a crosstalk between both pathways at the levels of the TCR complex and PKCθ. Putative interactions between FYN-TRAF3IP2 and the TCR complex and PKCθ warrant further exploration.

In all experiments, we observed comparably lower levels of the FYN-TRAF3IP2 fusion protein than mutant variants of the fusion protein. This could not be related to lower mRNA expression, because the fusion protein was expressed along with GFP from a bicistronic vector and we observed no decrease of the GFP

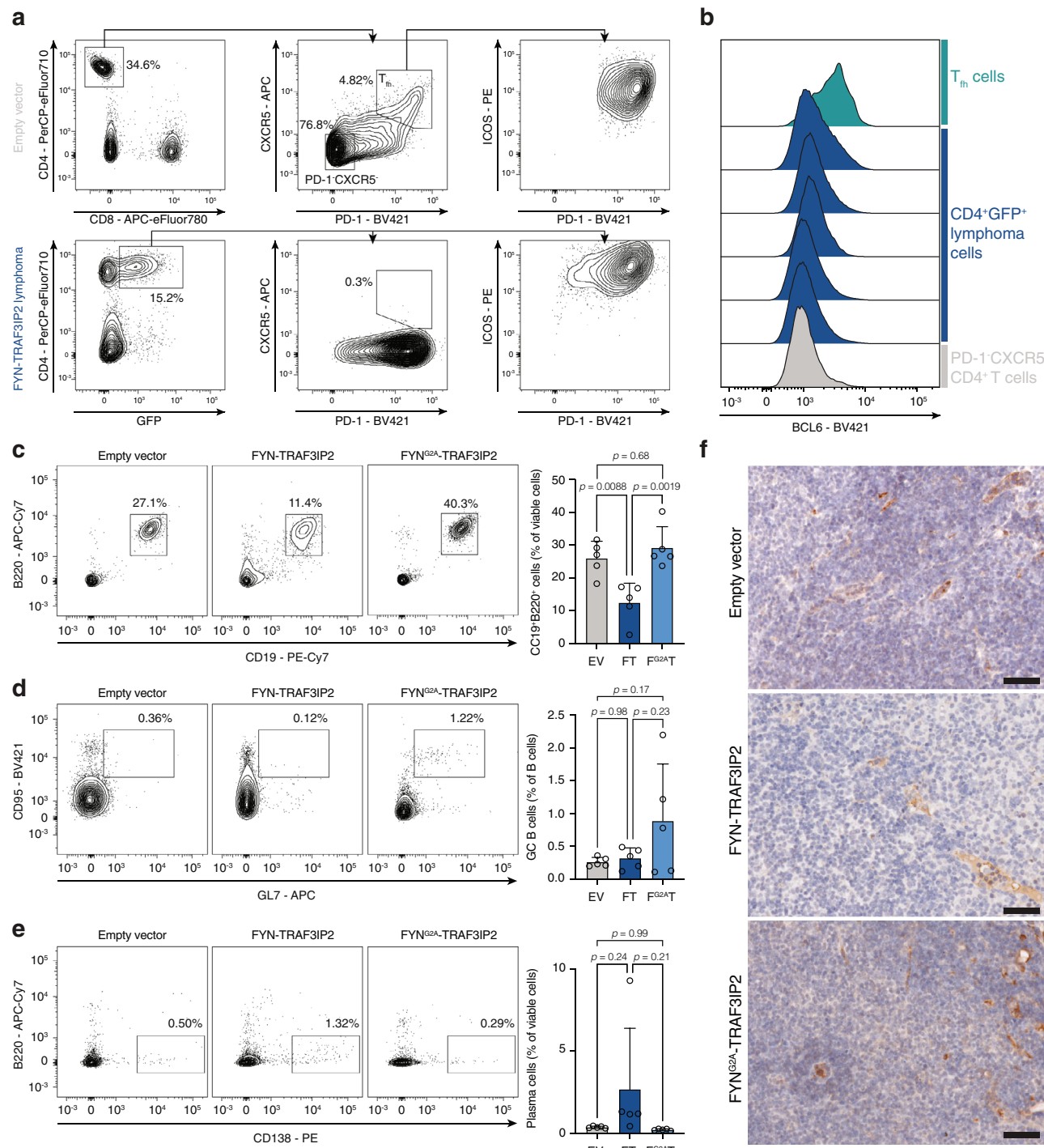

**Fig. 6 Characterization of *FYN-TRAF3IP2*-driven lymphomas in mice. a** Gating strategy for $T_{fh}$ cells on lymph node cell suspensions from mice with empty-vector-transduced cells (top). Representative flow cytometry plot for the immunophenotype of *FYN-TRAF3IP2*-driven lymphomas (bottom, $n = 9$). **b** Histograms for BCL6 protein in $T_{fh}$ cells (representative for $n = 2$), sorted viable CD4$^+$GFP$^+$ splenocytes from *FYN-TRAF3IP2*-driven lymphomas ($n = 5$) and PD-1$^-$CXCR5$^-$ CD4$^+$ T cells (representative for $n = 2$). Representative flow cytometry plots (left) and quantification (right) of CD19$^+$B220$^+$ B cells (**c**), CD95$^+$GL7$^+$ germinal center (GC) B cells (gated on CD19$^+$B220$^+$ B cells) (**d**) and CD19$^{lo-int}$B220$^{lo-int}$CD138$^+$ plasma cells (**e**) in lymph node suspensions from mice transplanted with empty-vector-transduced (EV) cells, *FYN-TRAF3IP2*-transduced (FT) cells or *FYN$^{G2A}$-TRAF3IP2*-transduced cells (F$^{G2A}$T). $n = 5$ mice per group. $p$ values were calculated with Tukey's post-hoc multiple comparisons test. **f** Representative pictures of CD31 immunohistochemistry on lymph node sections of mice transplanted with HSPC transduced with empty pMIG vector, pMIG-*FYN-TRAF3IP2* or pMIG-*FYN$^{G2A}$-TRAF3IP2*. $n = 5$ mice per group. Scalebars represent 50 µm. Data are represented as mean ± SD.

protein (Fig. 5g). These observations suggest that levels of the FYN-TRAF3IP2 fusion protein are likely controlled by post-transcriptional regulation and/or post-translational regulation. Persistent activation of TRAF3IP2 by IL-17 desensitizes cells

through degradation of TRAF3IP2 orchestrated by Skp1-cullin-1-F-box-type (SCF) E3 ubiquitin ligase complexes[41]. It is unknown whether levels of active FYN-TRAF3IP2 protein are regulated similarly by SCF-type E3 ubiquitin ligase complexes. Secondary

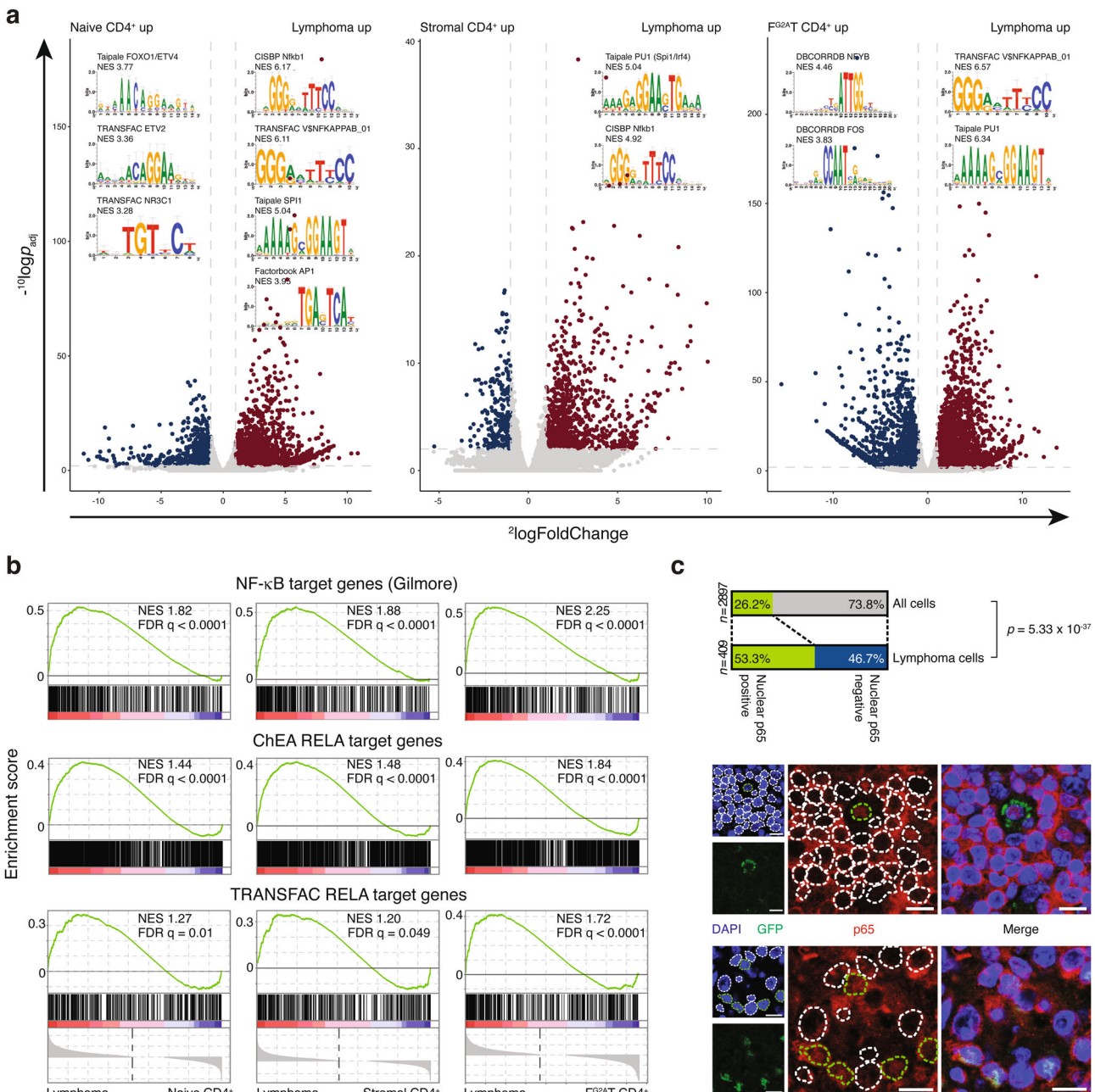

**Fig. 7 FYN-TRAF3IP2 activates canonical NF-κB signaling in vivo. a** Volcano plots of differentially expressed genes in CD4+GFP+ lymphoma cells versus naive CD4+ T cells (left), CD4+GFP− stromal T cells (middle) and CD4+GFP+ $FYN^{G2A}$-TRAF3IP2-expressing T cells (right). Genes significantly upregulated ($^2$logFoldChange > 1 and −$^{10}$logp$_{adj}$ > 2) in lymphoma cells are maroon-colored, genes significantly downregulated ($^2$logFoldChange < −1 and −$^{10}$logp$_{adj}$ > 2) in lymphoma cells are depicted in blue. Motifs from cis-regulatory features associated with differentially expressed genes are depicted. **b** Enrichment plots for a list of manually curated NF-κB target genes (top row), p65 target genes identified by ChIP-seq (middle row) and computationally predicted p65 target genes (bottom row) in lymphoma cells compared with naive CD4+ T cells (left column), CD4+GFP− stromal T cells (middle column) and CD4+GFP+ $FYN^{G2A}$-TRAF3IP2-expressing T cells (right column). **c** Quantification of cells with nuclear p65 positivity as a fraction of all cells in the lymphoma or as a fraction of GFP+ lymphoma cells in FYN-TRAF3IP2-induced lymphomas (top) and representative images (bottom). n = 5 mice. p value is derived from the cumulative distribution function of a hypergeometric distribution. Scalebars represent 10 μm.

lesions which impact FYN-TRAF3IP2 protein turnover, could enhance lymphomagenesis. The number of sequenced samples in this study did not allow statistical inference to identify events that co-occur with a *FYN-TRAF3IP2* gene fusion, but this will be of interest for future research.

Direct targeting of NF-κB signaling with IKK inhibitors has shown promise in preclinical models, but to date no IKK inhibitors are in clinical use. On one hand, they exhibit limited efficacy. On the other hand, they provoke prohibitive on-target

toxicities related, but not limited, to the ubiquitous activity of the NF-κB pathway in both the innate and adaptive immune system[42]. Therefore, we shifted our focus to downstream effectors of the NF-κB pathway. The ability of NF-κκB to promote cell survival through induction of target genes is well-described[26]. With the notable exception of *Mcl1*, we observed a global upregulation of pro-survival factors in murine lymphomas induced by *FYN-TRAF3IP2*. The expression of pro-apoptotic factors *Bax*, *Bak1*, BH3-only activators and BH3-only sensitizers was elevated

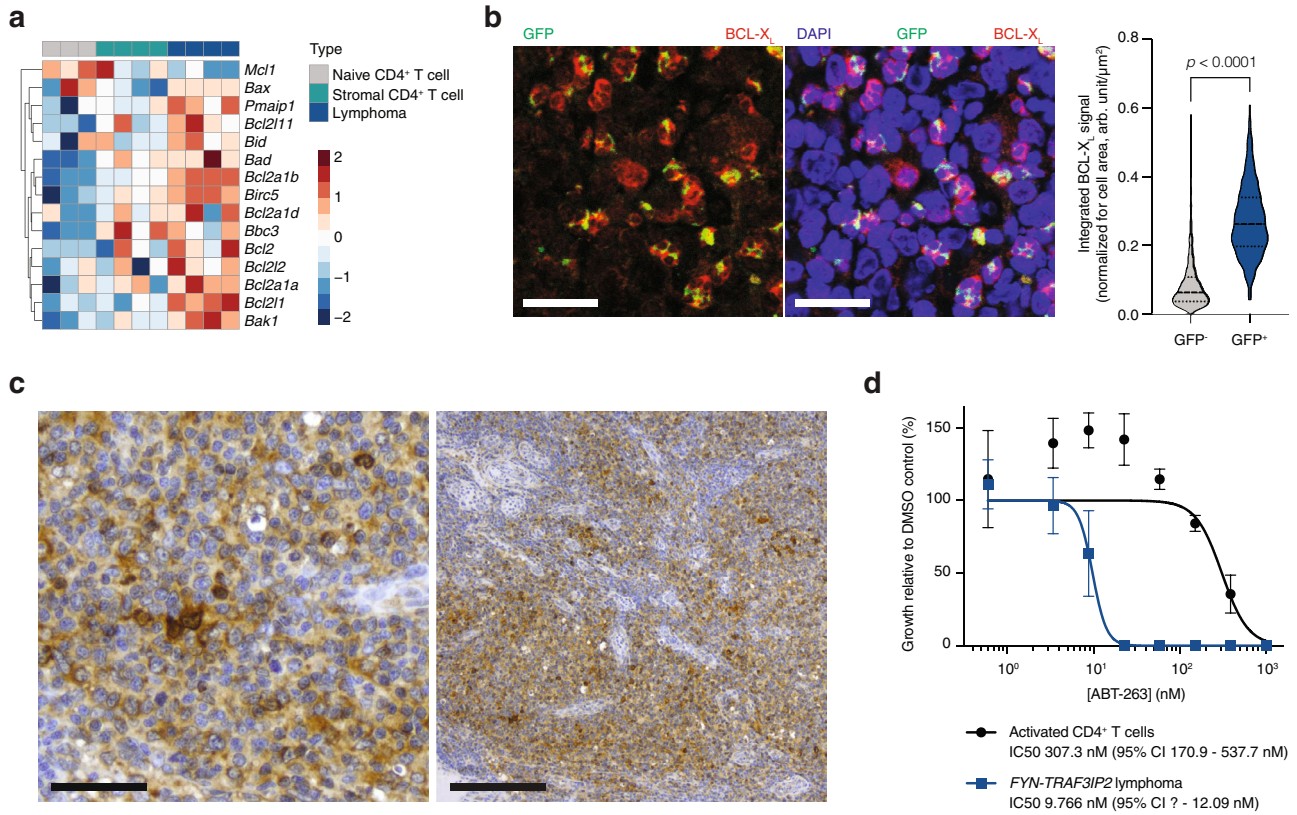

**Fig. 8 *FYN-TRAF3IP2*-driven NF-κB activation sensitizes lymphoma cells to inhibition of BCL-X$_L$/BCL2. a** Heatmap representation of relative transcript abundance of pro-survival and pro-apoptosis factors in naive CD4$^+$ T cells, CD4$^+$GFP$^-$ stromal T cells and *FYN-TRAF3IP2*-expressing lymphoma cells. The color scale represents the distribution for each row as mean ± SD. **b** Immunofluorescence images for BCL-X$_L$ and GFP in *FYN-TRAF3IP2*-driven murine lymphomas (right, n = 5 mice) and quantification of BCL-X$_L$ signal intensity in the lymphoma stromal cells (n = 4184 cells) and GFP$^+$ malignant cells (n = 510 cells). Scalebars represent 20 μm. Dashed lines represent the median and dotted lines represent the lower and upper quartile in the violin plots. *p* values were calculated with a two-sided Mann−Whitney test. **c** High (left) and low (right) magnification images of immunohistochemistry staining for BCL-X$_L$ on case PTCL2. Scalebars represent 50 μm (left) and 200 μm (right). **d** Growth inhibition of sorted *FYN-TRAF3IP2*-expressing lymphoma cells and sorted CD4$^+$ T cells after a 48-h treatment with ABT-263. n = 3 replicates per condition. Data are represented as mean ± SD.

to a lesser extent (Fig. 8a). *Bcl2l1* was one of the most significantly upregulated pro-survival genes. Expression of its gene product BCL-X$_L$ was confirmed in malignant cells in murine lymphomas and in a clinical specimen with a *FYN-TRAF3IP2* gene fusion. Given the increased expression of pro-apoptotic factors in *FYN-TRAF3IP2*-driven murine lymphomas, we reasoned that these cells are primed for apoptosis[43]. Indeed, *FYN-TRAF3IP2*-dependent lymphoma cells display exquisite sensitivity to the BCL-X$_L$, BCL2 and BCL-W (encoded by *Bcl2l2*) inhibitor ABT-263. ABT-263 is in clinical development for the treatment of myelofibrosis[44]. Thrombocytopenia is the most important side effect of BCL-X$_L$ inhibition, but is manageable with appropriate dosing. Moreover, a BCL-X$_L$ proteolysis targeting chimera is in development. This compound is more potent than ABT-263 at inhibiting BCL-X$_L$ while at the same time it minimizes toxicity to thrombocytes[45]. Considering the importance of NF-κB in the regulation of cell survival, these results provide a strong rationale to target BCL-X$_L$ in *FYN-TRAF3IP2*-driven lymphomas.

In addition to the recurrent *FYN-TRAF3IP2* fusion, we identified a *KHDRBS1-LCK* fusion, which represents a prototypic fusion kinase. In resting T cells, a significant fraction of LCK is catalytically active[46] through autophosphorylation of tyrosine 394 in the LCK activation loop[47]. The KHDRBS1-LCK fusion protein bears the KHDRBS1 dimerization domains[17], which could enhance LCK transphosphorylation and activation. Indeed, constitutive dimerization of a tyrosine kinase domain by fusion with a self-associating partner protein is a recurrent mechanism in

oncogenic fusion proteins and was able to activate LCK in an insertion mutagenesis screen[48]. We show that KHDRBS1-LCK kinase activates different branches of the TCR signaling cascade. We provide evidence that targeting LCK kinase activity with dasatinib has potent inhibitory effects on KHDRBS1-LCK-driven signaling, paving the way for therapeutic intervention in a clinical setting. Early-stage clinical trials suggest activity of dasatinib in PTCL-NOS[49] and TFH-related PTCL[50]. With the identification of the *KHDRBS1-LCK* gene fusion, we provide an additional biological basis to repurpose dasatinib for the treatment of PTCL-NOS.

Together, our data provide genetic and biologic insights in PTCL-NOS and PTCL-TFH as well as a rationale for the design of novel treatment regimens for PTCL cases with these fusion genes. Overall, these data warrant further study of abnormal TCR signaling in PTCL.

## Methods
**Patient samples.** Patient samples for the discovery cohort were collected retrospectively from the tumor banks of the University Hospitals Leuven and the CHU Mont-Godinne and prospectively in the University Hospitals Leuven. For prospectively obtained samples, we obtained informed consent from all patients. All cases were reviewed by two hematopathologists (LM and TT).

Patient samples for the validation cohort were obtained from the T-cell lymphoma biobank (TENOMIC) of the Lymphoma Study Association (LYSA) and were reviewed by two hematopathologists (PG en LdL). The study was approved by the Ethics Committee UZ/KU Leuven (S62100).

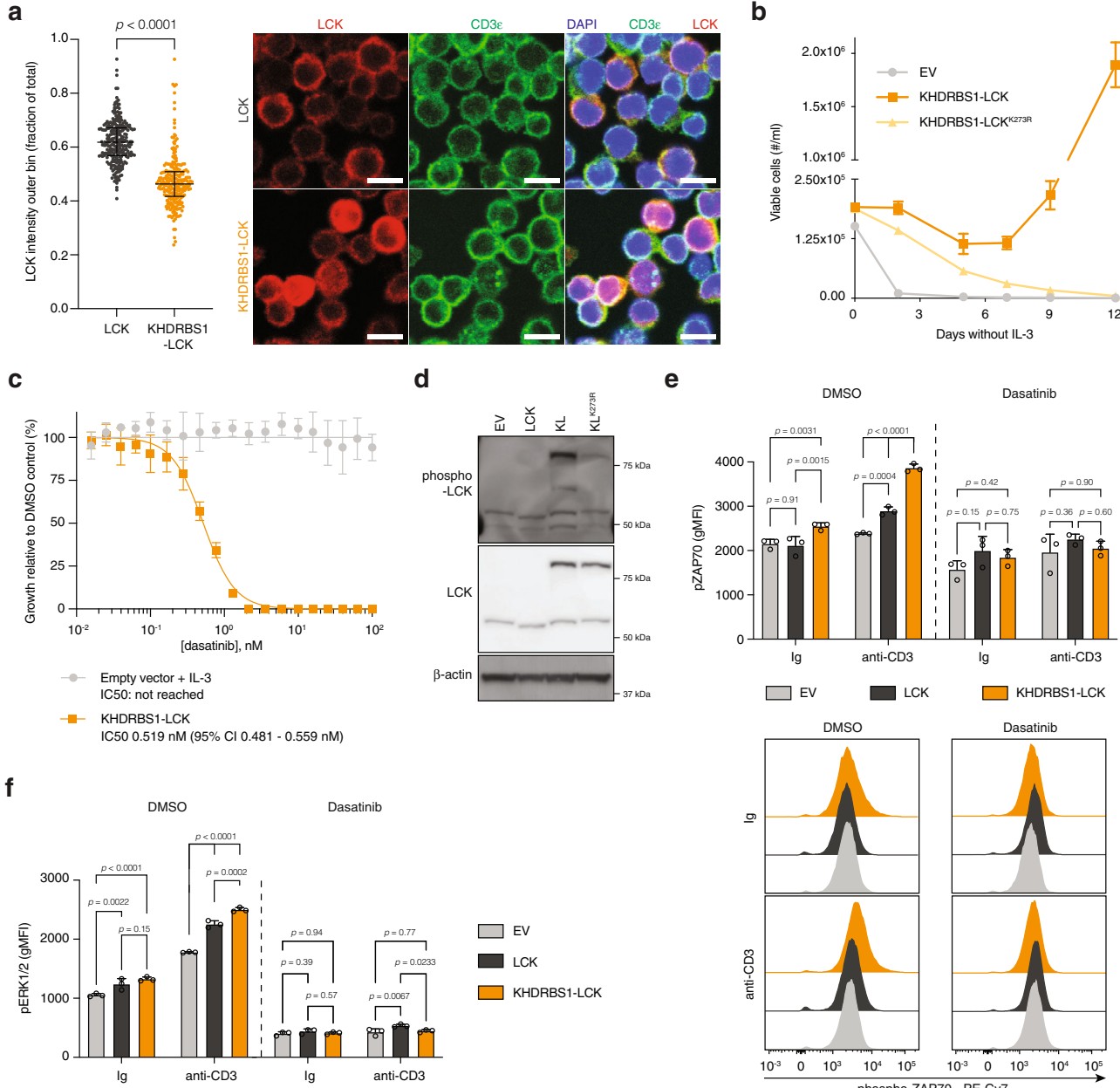

**Fig. 9 KHDRBS1-LCK drives chronic active TCR signaling. a** Quantification (left) and representative immunofluorescence images (right) of LCK and CD3ε staining on primary T cells with ectopic expression of *LCK* ($n = 268$ cells) or *KHDRBS1-LCK* ($n = 242$ cells). Each dot represents a cell. Horizontal line and whiskers represent median and interquartile range respectively. *p* values were calculated with a two-sided Mann−Whitney test. Scalebars represent 10 μm. **b** Outgrowth of Ba/F3 cells transduced with empty pMIG vector, pMIG-*KHDRBS1-LCK* or pMIG-*KHDRBS1-LCK*^K273R^ after withdrawal of IL-3. $n = 3$ biological replicates per condition. **c** Growth inhibition of Ba/F3 cells transformed by *KHDRBS1-LCK* or Ba/F3 cells transduced with empty pMIG vector after a 24-h treatment with dasatinib. $n = 3$ replicates per condition. **d** Western blot for total LCK and phospho-LCK (Tyr394) in Jurkat cells transduced with empty pMIG vector, pMIG-*LCK*, pMIG-*KHDRBS1-LCK* or pMIG-*KHDRBS1-LCK*^K273R^. **e** Quantification (top) and representative flow cytometry histograms (bottom) for phospho-ZAP70 in primary T cells treated with control immunoglobulin (Ig) or agonistic anti-CD3ε antibody in the absence or presence of dasatinib. $n = 3$ biological replicates per condition. *p* values were calculated with Tukey's post-hoc multiple comparisons test. **f** Intracellular flow cytometry quantification of phospho-ERK1/2 levels in primary T cells treated with control immunoglobulin (Ig) or agonistic anti-CD3ε antibody in the absence or presence of dasatinib. $n = 3$ biological replicates per condition. *p* values were calculated with Tukey's post-hoc multiple comparisons test. All data are represented as mean ± SD unless stated otherwise.

**Mice**. We used 6- to 10-week-old C57BL/6J mice for the isolation of primary T cells and bone marrow transplant experiments. Mouse experiments were approved and supervised by the KU Leuven ethical committee and conducted according to EU legislation (Directive 2010/63/EU).

Mice were housed in individually ventilated cages with a temperature between 18 and 23 °C and humidity between 40 and 60%. No more than 5 mice were housed in a single cage. The room had a programmed 12-h light and 12-h dark cycle.

**RNA extraction, cDNA synthesis and qRT-PCR**. RNA from clinical samples was extracted from 4 cryosections, each 10 μm thick. Briefly, sections were resuspended in Trizol (Thermo Fisher Scientific). After addition of chloroform, RNA was precipitated from the aqueous phase with 100% ethanol. RNA was washed and eluted with the RNeasy Mini Kit (Qiagen) according to the manufacturer's instructions. Concentrations and purity were measured with the NanoDrop 2000 (Fisher Scientific). RNA integrity was measured with the Bioanalyzer 2100 system using the RNA 6000 Nano Kit (both from Agilent). Samples with an RNA integrity

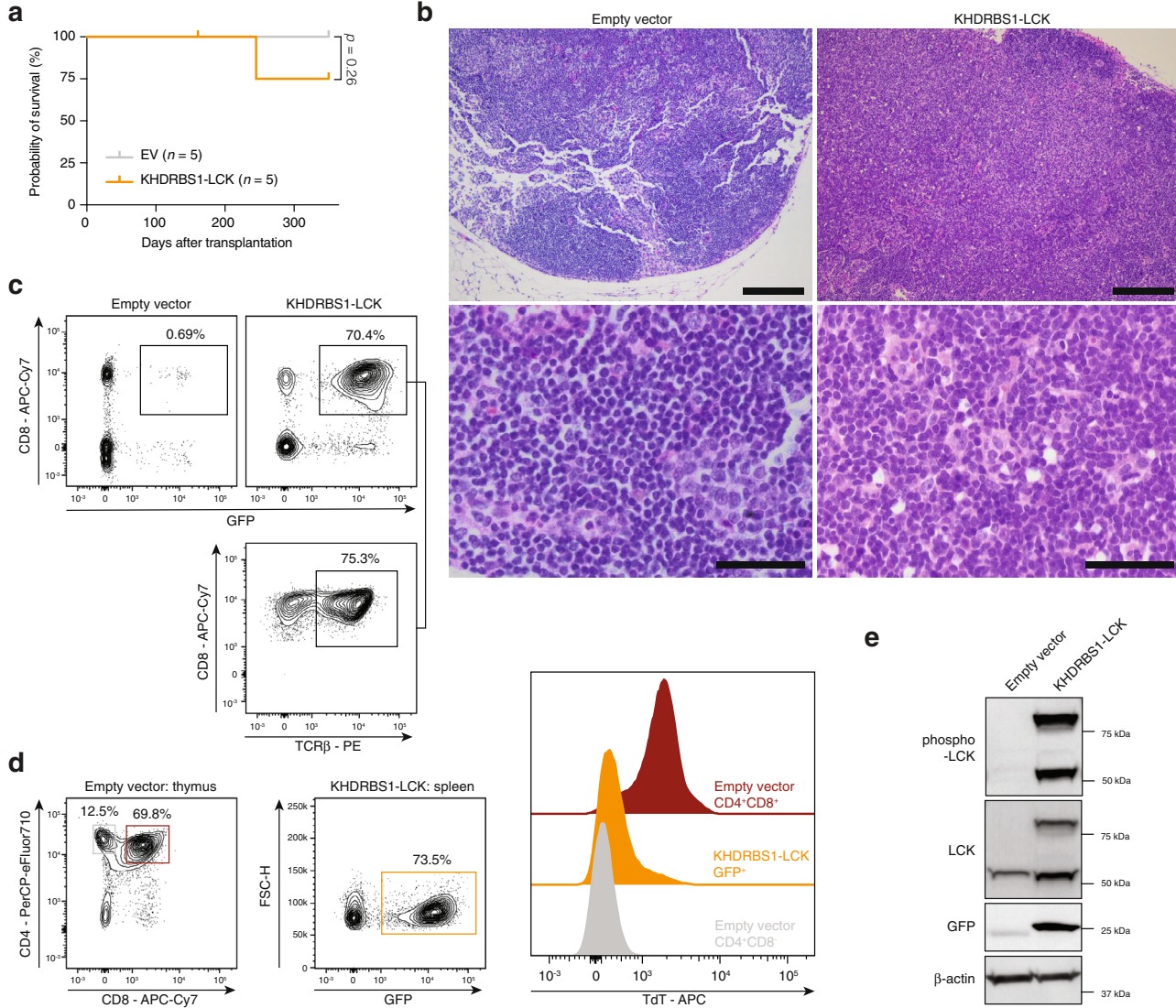

**Fig. 10 Characterization of *KHDRBS1-LCK*-driven PTCL in mice. a** Kaplan–Meier survival curve after transplantation with HSPC transduced with empty pMIG vector ($n = 5$) or pMIG-*KHDRBS1-LCK* ($n = 5$). Log-rank $p$ value was obtained from a two-sided Chi-square test. **b** Representative H&E stains of lymph nodes from mice who were transplanted with HSPC transduced with empty vector ($n = 5$) or *KHDRBS1-LCK* ($n = 1$). Low magnification images are in the top row, scalebars represent 200 μm. High magnification images are in the bottom row, scalebars represent 50 μm. **c** Representative flow cytometry plots for cell suspensions from the lymph nodes of empty vector mice (representative for $n = 5$) and the *KHDRBS1-LCK* mouse ($n = 1$). **d** Gating strategy for CD4⁺CD8⁺ double positive cells and CD4⁺CD8⁻ single positive cells on thymic cell suspensions from mice with empty-vector-transduced cells and GFP⁺ cells in *KHDRBS1-LCK* induced PTCL (left). Quantification of intracellular TdT with intracellular flow cytometry. **e** Western blot for phospho-LCK (Tyr394) and total LCK in spleen lysates from mice transduced with HSPC transduced with empty vector or *KHDRBS1-LCK*-induced PTCL.

number >8 were retained for RNA sequencing. cDNA synthesis was performed using SuperScript IV (Thermo Fisher Scientific). To detect fusion transcripts, cDNA was amplified with GoTaq polymerase (Promega) using the primers listed in Supplementary Table 3.

Mouse tissue was disrupted in Trizol (Thermo Fisher Scientific) with the TissueLyser LT (Qiagen) at 50 Hz for 4 min. Supernatants were collected. After addition of chloroform, RNA was precipitated from the aqueous phase with 100% ethanol. RNA was washed and eluted with the RNeasy Mini Kit (Qiagen) according to the manufacturer's instructions. Concentrations and purity were measured with the NanoDrop 2000 (Fisher Scientific). cDNA synthesis was performed using GoScript (Promega). T cell receptor clonality was verified on cDNA with GoTaq polymerase (Promega) using the primers listed in Supplementary Table 3. PCR products were loaded on the QIAxcel System (Qiagen).

RNA was extracted from cultured cells using the Maxwell RSC simplyRNA Cells purification kit (Promega). cDNA synthesis was performed using GoScript (Promega) and qRT-PCR was performed using the GoTaq qRT-PCR master mix (Promega) on the ViiA7 Real Time PCR system (Applied Biosystem). Primers used for qRT-PCR are listed in Supplementary Table 3. qRT-PCR data were analyzed with the geNorm algorithm in the R package SLqPCR v1.54.0 (https://doi.org/10.18129/B9.bioc.SLqPCR) in R version 4.0.

**RNA sequencing.** For clinical samples, libraries were made with the TruSeq Stranded mRNA Library Prep Kit (Illumina). We used 1 μg RNA as input. Libraries were prepared per manufacturer's instructions. Libraries were pooled and sequenced on an S2 flow cell on a NovaSeq 6000 system (both Illumina) in 2 × 100 bp mode to an average depth of 200 million reads per sample.

For mouse cells, libraries were made with the QuantSeq 3' mRNA-Seq Library Prep Kit (Lexogen). We used 20–100 ng RNA as input. Libraries were prepared per manufacturer's instructions. Libraries were pooled and sequenced on a HiSeq 4000 system (Illumina) in 1 × 50 bp mode.

**Bioinformatics.** Raw sequencing data from the Illumina sequencer was demultiplexed and converted to the fastq file format with Illumina's bcl2fastq v2.20 software. For anaplastic large cell lymphoma samples[12] and normal lymph nodes[51], we used data from publicly accessible repositories. Quality of fastq files was assessed with FastQC v0.11.9. RNA-sequencing data was first cleaned (i.e., removal of adapters and low-quality reads) with fastq-MCF (ea-utils v1.1.2), again followed by quality control with FastQC. Chimeric reads were identified with FusionCatcher[52] v1.0. Reads were aligned with HISAT2[53] v2.1.0 to their respective genomes, either Homo Sapiens (GRCh38/hg38) or Mus Musculus (mm10) and

 **15**

coordinate-sorted with SAMtools v1.10[54]. HTSeq-count[55] v.0.9.1 was used to count the number of reads per gene. Differential gene expression analysis was performed with the R-package DESeq2[56] v1.24.0, using R version 4.0. Gene set enrichment analysis was performed with the Broad Institute GSEA software[57] v4.1.0. Analysis of cis-regulatory features associated with co-expressed genes was performed with i-cisTarget[58].

**Genomic DNA**. Genomic DNA from clinical samples was extracted with the DNEasy kit (Qiagen) according to the manufacturer's instructions. PCR on genomic DNA was performed with Q5 polymerase (New England Biolabs) and the primers listed in Supplementary Table 3.

**Plasmids and vectors**. RNA from patient samples and Jurkat cells were retro-transcribed with SuperScript IV reverse transcriptase (Thermo Fisher Scientific) using oligo(dT)15 primers (Promega). $FYN^{1-232}$, $TRAF3IP2$, $FYN$-$TRAF3IP2$, $FYN^{G2A}$-$TRAF3IP2$, $LCK$ and $KHDRBS1$-$LCK$ were cloned from cDNA with Q5 polymerase (New England Biolabs) and the primers listed in Supplementary Table 3. Blunt-ended cDNA amplicons were ligated in the pJET1.2 cloning vector (Thermo Fisher Scientific) and verified with Sanger sequencing (Eurofins Genomics). Validated coding sequences were cloned from the pJET1.2 cloning vector to the multiple cloning site of the pMSCV-IRES-GFP II retroviral expression vector (Addgene #52107) or pMSCV-IRES-mCherry retroviral expression vector (Addgene #52114) and verified with Sanger sequencing (Eurofins Genomics).

$FYN$-$TRAF3IP2^{ΔT6}$ and $KHDRBS1$-$LCK^{K273R}$ were generated with the Q5 site-directed mutagenesis kit (New England Biolabs) according to manufacturer's instructions with primers listed in Supplementary Table 3.

The pGreenfire1-NF-κB luciferase/GFP reporter lentiviral transfer plasmid was ordered from System Biosciences (TR012PA-1).

**Cell culture**. Ba/F3 cells were cultured in RPMI 1640 supplemented with 10% fetal bovine serum (FBS) and IL-3 (1 ng/ml; Peprotech). For transformation assays, cells were washed 3 times with PBS to remove all IL-3 and resuspended in RPMI 1640 supplemented with 10% FBS at a density of $10^5 - 1.5 \times 10^5$ cells/ml. Cell growth and GFP percentage were monitored with a MACSQuant Vyb cytometer (Miltenyi).

Jurkat cells were cultured in RPMI 1640 with 20% FBS. For anti-CD3 stimulation, Jurkat cells were resuspended in chilled PBS with 1 µg/ml anti-CD3 (OKT3, Biolegend) at a density of $10^6$ cells/ml and incubated on ice for 20 min. Cells were then resuspended in full medium with 10 µg/ml goat anti-mouse IgG (Biolegend) at cell densities of $10^7$/ml. After a 10-min incubation at room temperature, cells were diluted to a final concentration of $10^6$/ml in full medium and incubated overnight. For phorbol 12-myristate 13-acetate (PMA) stimulation, Jurkat NF-κB luciferase reporter cells were seeded in 200 µl RPMI 1640 with 10% FBS at a density of $10^6$ cells/ml with or without PMA/ionomycin (50 nM PMA, 13.4 µM ionomycin; Thermo Fisher Scientific). After 6 h the treatment was stopped.

Primary T cells were isolated from spleens of wild type C57BL/6J mice with the MojoSort Mouse CD4 Naive T cell isolation kit or Mouse CD3 T Cell Isolation Kit (both BioLegend). Briefly, spleens were smashed on a 40 µm cell strainer. After red blood cell lysis, T cells were purified according to the manufacturer's instructions. Purified cells were resuspended in T cell medium (RPMI 1640 supplemented with 10% FBS, 2 mM glutamine, non-essential amino acids solution (Gibco), 1 mM sodium pyruvate, 50 µM 2-mercaptoethanol, Primocin (Invivogen) and IL-2 (10 ng/ml; Peprotech)). T cells were activated on anti-CD3 coated plates (clone 145-2C11, 2 µg/ml in PBS, BioLegend) in T cell medium supplemented with anti-CD28 (clone 37.51, 2 µg/ml, BioLegend) for 24 h. After 24 h, T cells were transferred to tissue culture-treated plates and cultured in T cell medium. T cells were split daily, to maintain a density of $1$-$2 \times 10^6$ cells/ml. After 7 days, T cells were briefly restimulated with soluble anti-CD3 (clone 145-2C11, 0.5 µg/ml, BioLegend) and anti-CD28 (clone 37.51, 0.5 µg/ml, BioLegend). For IL-17 stimulation, cells were cultured overnight in T cell medium supplemented with IL-17A (100 ng/µl; Peprotech). For anti-CD3 stimulation, T cells were resuspended in RPMI 1640 without FBS with 10 µg/ml anti-CD3 (145-2C11, Biolegend) or 10 µg/ml Armenian hamster IgG isotype control (eBioscience) at a density of $10^7$ cells/ml and incubated on ice for 15 min in polypropylene FACS tubes (BD). For stimulation, tubes were transferred to a water bath at 37 °C. Treatment was stopped by the addition of fixation buffer (after 1 min for phospho-ZAP70, after 15 min for all other analyses). For experiments with dasatinib, T cells were incubated in RPMI 1640 with 50 nM dasatinib (Selleckchem) or DMSO at 37 °C for 15 min prior to stimulation with anti-CD3 antibody.

**CRISPR/Cas9 genome editing**. To generate $CARD11$ knock-out Jurkat cells, we cloned oligos complementary to the target site (Supplementary Table 3) in the pX330 vector (Addgene #42230) after digestion with BbsI (Thermo Fisher Scientific). We electroporated the pX330 plasmid in Jurkat cells in serum-free RPMI 1640 medium using the Gene Pulser Xcell™ system (Biorad). Immediately after electroporation, cells were transferred to 2 ml pre-warmed RPMI 1640 medium supplemented with 20% FBS. After 48 h, single cells were cultured in 96-well plates. Clones that grew out, were screened with Western blot. All experiments with

$CARD11$ knock-out Jurkat cells were reproduced in clones 4 and 6 (Supplementary Fig. 3d).

**Virus production and transduction**. Ecotropic retrovirus was produced In 293T cells transfected with the appropriate retroviral expression vector and the pIK6.1MCV.ecopac packaging plasmid using GeneJuice transfection reagent (Merck Millipore). 24 h after transfection, the medium was changed to RPMI 1640 with 10% FBS. 24 h later, the supernatant with viral particles was harvested. One ml of viral supernatant was added to $10^6$ Ba/F3 cells in 1 ml of medium and cells were transduced overnight in the presence of polybrene (8 µg/ml). Primary T cells were transduced as reported previously[59].

VSV-G pseudotyped retrovirus was produced in 293T cells transfected with retroviral expression vector, gag-pol (Addgene #14887) and pCMV-VSV-G plasmid (Addgene #8454) using Genejuice (Merck Millipore). After 6 h, the medium was changed to RPMI 1640 with 10% FBS. 48 h after transfection, supernatant carrying viral particles was harvested. Jurkat cells were transduced on retronectin-coated plates (Takara) by spinfection.

Luciferase/GFP dual NF-κB reporter lentivirus was produced in 293T cells transfected with pGreenFire1-NF-κB luciferase/GFP plasmid, psPAX2 (Addgene #12260) and pCMV-VSV-G plasmid (Addgene #8454) using GeneJuice (Merck Millipore). After 6 h, the medium was changed to RPMI 1640 with 20% FBS. Supernatant carrying the viral particles was harvested 48 h after transfection and concentrated on a 20% sucrose gradient by ultracentrifugation at $112,500 \times g$ for 2 h. Jurkat cells were transduced on retronectin-coated plates (Takara) by spinfection (1 h at $2000 \times g$).

**Murine bone marrow transplantation**. C57BL/6J mice were purchased from Charles River Laboratories. Bone marrow cells were harvested from femur and tibia of male mice. Lineage negative cells were enriched (EasySep Mouse Hematopoietic Progenitor Cell Isolation Kit, STEMCELL Technologies) and cultured overnight in RPMI 1640 supplemented with 20% FBS, IL-3 (10 ng/ml; Peprotech), IL-6 (10 ng/ml; Peprotech), SCF (50 ng/ml; Peprotech), and Primocin (Invivogen). The following day, cells were transduced by spinoculation (90 min at $2000 \times g$) with viral supernatant and 8 µg/mL polybrene and syngeneic female recipient mice were sublethally irradiated (5 Gy). The following day, the cells were washed in PBS and injected ($1 \times 10^6$ cells/ 0.3 ml) into the lateral tail vein of sublethally irradiated syngeneic female recipient mice. Mice were housed in individually ventilated cages and monitored daily.

**Immunoblotting and co-immunoprecipitation**. Cells were lysed in Cell Lysis buffer (Cell Signaling). Tissues were homogenized with a TissueLyserLT (Qiagen). Cytoplasmic and membrane fractions were obtained with the Mem-PER Plus Membrane Protein Extraction Kit (Thermo Fisher Scientific) according to the manufacturer's instructions and the addition of additional wash steps with PBS.

For immunoprecipitation of TRAF6, 50 µl of Dynabeads Protein G (Thermo Fisher Scientific) were pre-coated with anti-TRAF6 antibody (Cell Signaling, #8028S) diluted 1:50 per IP and IP was performed according to the manufacturer's instructions. For immunoprecipitation of TRAF3IP2, 50 µl of Dynabeads sheep anti-rat IgG (Thermo Fisher Scientific) were mixed with lysates, 10 µg/ml anti-TRAF3IP2 antibody (Thermo Fisher Scientific, #14-4040-82) and 5% normal donkey serum. The beads-lysate mixtures were incubated over night at 4 °C with gentle rotation. After thorough washing, protein complexes were eluted in denaturing NuPAGE LDS sample buffer (Thermo Fisher Scientific) and loaded on the gel.

The proteins were separated on NuPAGE NOVEX Bis-Tris 4–12% gels (Life Technologies) and transferred to PVDF membranes. Subsequent Western blot analysis was performed using antibodies listed in Supplementary Table 4, diluted in TBS with 0.1% Tween 20 and 5% non-fat milk. Western blot detection was performed with secondary antibodies conjugated with horseradish peroxidase (Cytiva) diluted 1:5000 in TBS with 0.1% Tween 20 and 5% non-fat milk. Bands were visualized using a cooled charge-coupled device camera system (ImageQuant LAS-4000; GE Health Care).

**Drug treatment**. Ba/F3 cells, MACS-sorted naive CD4$^+$ T cells or FACS-sorted CD4$^+$GFP$^+$ lymphoma cells were seeded in 96-well plates. Ba/F3 cells were cultured in RPMI 1640 with 10% FBS with or without IL-3. T cells were cultured in T cell medium. Compounds were dissolved in DMSO and added in a randomized fashion using a D300e digital dispenser (Tecan). Cell proliferation was measured at baseline and after 24 h (Ba/F3) or 48 h (lymphoma and T cells) using the ATPlite luminescence system (PerkinElmer) on a Victor multilabel plate reader (PerkinElmer). After subtraction of the baseline signal, results were normalized to the DMSO control signal.

**Luciferase**. Jurkat NF-κB luciferase reporter cells were seeded in 200 µl RPMI 1640 with 10% FBS at a density of $10^6$ cells/ml with or without phorbol 12-myristate 13-acetate (PMA)/ionomycin (50 nM PMA, 13.4 µM ionomycin; Thermo Fisher Scientific). After 6 h, cells were washed in PBS and lysed in 30 µl passive lysis buffer (Promega) with gentle agitation. After 20 min, 20 µl of lysate was transferred to a new plate and loaded on a Victor multilabel plate reader (PerkinElmer). With the

dispenser, 100 μl of Luciferase Assay Reagent II (Promega) was added and luci-ferase signal was measured well per well.

**Flow cytometry**. For surface staining, single cell suspensions were blocked with unconjugated CD16/32 antibody and stained with Fixable viability dye in staining buffer (PBS with 2% FBS) for 10 min protected from light at room temperature. Next, cells were washed with stain buffer. Cells were resuspended in staining buffer with the appropriate antibody dilutions for 30 min at 4 °C protected from light. For the detection of cytosolic proteins, cells were fixed with room-temperature IC fixation buffer and permeabilized with eBioscience Permeabilization buffer (both from Thermo Fisher Scientific). For the detection of phosphorylated cytosolic proteins, cells were fixed with room-temperature IC fixation buffer and permeabilized with ice-cold methanol. For the detection of phosphorylated transcription factors, cells were pre-pared with the Transcription Factor Buffer set (BD Biosciences) according to the manufacturer's instructions. For the detection of BCL6, viable CD4+GFP+ lymphoma cells were first sorted (MA900 cell sorter, Sony Biotechnology). For the detection of TdT, viable GFP+ lymphoma cells were first sorted (S3 cell sorter, Biorad). Sorted cells were processed with the FoxP3 transcription factor staining buffer set (Thermo Fisher Scientific) according to the manufacturer's protocol. Antibodies and dilutions are listed in Supplementary Table 5. Data were acquired on Fortessa X-20, FACS-Canto II or FACSVerse cell analyzers (all from BD Biosciences). Data were analyzed with FlowJo version 10.6 (BD Biosciences).

CD4+GFP+ lymphoma cells, CD4+GFP− stromal cells from *FYN-TRAF3IP2*-driven lymphomas and CD4+GFP+ *FYN^{G2A}-TRAF3IP2*-expressing cells were sorted with an S3 cell sorter (Biorad) prior to RNA-seq.

Gating strategies are summarized in Supplementary Fig. 7.

**Histology and immunofluorescence staining**. For the membrane staining of FYN-TRAF3IP2 in 293T cells, we coated cover slips with poly-L-lysine. 293T cells were seeded on top of the cover slips in RPMI 1640 with 10% FBS. After 6 h, cells were transfected with empty pMIG vector or pMIG constructs encoding *TRA-F3IP2*, *FYN-TRAF3IP2* or *FYN^{G2A}-TRAF3IP2* using GeneJuice transfection reagent (Merck Millipore). After 12 h, the medium was replaced with RPMI 1640 with 10% FBS. 36 h later, cells were fixed for 20 min with 4% paraformaldehyde in PBS at room temperature. Next, cells were blocked in staining buffer (4% BSA, 0.1% digitonin in PBS) for 2 h at room temperature. After blocking, cover slips were carefully removed from the plates and stained over night at 4 °C with primary anti-TRAF3IP2 antibody (Thermo Fisher Scientific, #14-4040-82) diluted 1:200 in staining buffer. The next day, cover slips were washed with staining buffer followed by staining with Alexa Fluor 647-conjugated goat anti-rat antibody (Thermo Fisher Scientific, A-21247) diluted 1:500 in staining buffer for 1 h at room temperature. Finally, cover slips were washed with PBS, counterstained with DAPI and mounted with ProLong Gold Antifade mountant (Thermo Fisher Scientific, P36930).

For the membrane staining of TRAF3IP2 and LCK on GFP-sorted primary T cells, 400,000 sorted cells were loaded in cytospin columns (Hettich) and spun down for 6 min at 400 × *g* on glass slides. Cells were fixed with 4% paraformaldehyde in PBS for 20 min at room temperature. Further processing was identical to the membrane staining protocol in 293T cells. For LCK and CD3E staining, cells were incubated over night at 4 °C with anti-LCK antibody (Cell Signaling, 2787S) diluted 1:200 and anti-CD3E antibody (BioLegend, #100223) diluted 1:500 in staining buffer. After washing, cells were stained with Alexa Fluor 555-conjugated goat anti-rat antibody (Themo Fisher Scientific, A-21434) and Alexa Fluor 647-conjugated donkey anti-rabbit antibody (Thermo Fisher Scientific, A-31573), both diluted 1:500 in staining buffer, for 1 h at room temperature.

Mouse tissues were fixed over night at 4 °C in 10% neutral buffered formalin (Sigma). Tissues were transferred to ethanol 70% followed by paraffin embedding. 4 μm sections were mounted on glass slides. Antigen retrieval was performed as previously described[60].

For immunohistochemistry, endogenous peroxidases were blocked with Dako REAL peroxidase-blocking solution (Agilent). Slides were blocked for 30 min at room temperature with 10% donkey serum diluted in PBS. Slides were incubated over night at 4 °C with anti-GFP (Cell Signaling, #2956S) diluted 1:200, anti-CD31 (BD Biosciences, 550274) diluted 1:50 or anti-BCL-XL (Cell Signaling, #2764S) diluted 1:200 in Dako REAL antibody diluent (Agilent) supplemented with 5% donkey serum. The next day, slides were washed with Dako Omnis wash buffer (Agilent) and stained with Dako EnVision+ Dual Link System (DAB+) (Agilent).

For immunofluorescence staining, slides were blocked for 30 min at room temperature with 10% donkey serum diluted in PBS. For GFP and TRAF3IP2 double immunofluorescence staining, slides were incubated over night at 4 °C with anti-GFP (Cell Signaling, #2956S) diluted 1:200 and anti-TRAF3IP2 (Thermo Fisher Scientific, #14-4040-82) diluted 1:200 in Dako Real antibody diluent (Agilent) supplemented with 5% donkey serum. The next day, slides were washed with Dako Omnis wash buffer (Agilent) and stained with Rhodamine Red-X-conjugated donkey anti-rabbit antibody (Jackson ImmunoResearch, 711-295-152) diluted 1:300 and Alexa Fluor 647-conjugated goat anti-rat antibody (Thermo Fisher Scientific, A-21247) diluted 1:500 in Dako REAL antibody diluent (Agilent) for 1 h at room temperature. Because we observed complete correlation between the GFP channel and the Rhodamine Red-X channel, the native GFP signal was acquired for other immunofluorescence stainings. Slides were incubated over night at 4 °C with anti-NF-κB p65 (Cell Signaling, #8242S) diluted 1:200, anti-RELB

(Cell Signaling, #10544S) diluted 1:400 or anti-BCL-XL (Cell Signaling, #2764S) diluted 1:200 in Dako REAL antibody diluent (Agilent) supplemented with 5% donkey serum. The next day, slides were washed with Dako Omnis wash buffer (Agilent) and stained with Alexa Fluor 647-conjugated donkey anti-rabbit antibody (Thermo Fisher Scientific, A-31573) diluted 1:500 in Dako REAL antibody diluent (Agilent) for 1 h at room temperature. Slides were then washed, followed by a counterstain with DAPI. Slides were washed with PBS and mounted with ProLong Gold Antifade mountant (Thermo Fisher Scientific, P36930).

**Imaging**. Immunofluorescence images were acquired using the SP8 X confocal microscope (Leica). H&E slides and immunohistochemistry slides were imaged with a conventional light microscope (Leica) or a Vectra Polaris slide scanner (Akoya Biosciences).

**Image analysis**. All image analysis was done with CellProfiler v4.0.6 software[61]. For all analyses, images were segmented based on the DAPI channel. For analysis of membrane staining, cells were divided in bins and the integrated signal intensity was measured in every bin and expressed as a fraction of integrated signal intensity over the entire cell mask. For the analysis of nuclear RELA accumulation, cells were classified as GFP-positive or GFP-negative based on the GFP signal in each indi-vidual segment. Next, we measured the NF-κB signal intensity within the DAPI masked region, shrunk by 2 pixels to avoid interference from the cytoplasmic signal at the nuclear rim. We used the lowest quartile intensity signal to classify cells based on their nuclear NF-κB signal. For the analysis of nuclear RELB accumu-lation, cells were classified as RELB-positive or RELB-negative based on the RELB signal in each individual segment. Subsequently RELB-positive cells were classified as GFP-positive or -negative. RELB integrated signal intensity was measured over the nuclear mask and the cell mask after subtraction of the background signal (mean RELB signal in the inverted cell mask). For the quantification of BCL-XL, BCL-XL integrated signal intensity was measured over the cell mask after sub-traction of the background signal and normalized to cell area.

**Statistics and reproducibility**. Statistical analysis and graphical representation were performed with Prism version 9 (GraphPad) and the R language (version 4.0). The Western blot in Fig. 10e was performed once. Figures 2b, c, e–g, 3a, c–f, 4a–f, 5g, 8d, 9a, d–f and Supplementary Figs. 2a, 3a–e, 6a are representative of 2 independent experiments with similar results. Figures 2a, d, 3b, 9b, c are representative of 3 independent experiments with similar results. Unless stated differently, all graphs use the mean as a measure of central tendency and the standard deviation as a measure of variation. For in vitro assays, we analyzed at least 3 biological replicates per condition. For qRT-PCR data, every data point represents the average of 3 technical replicates of a single biological replicate. Boxplots were made with the R package ggplot2 v3.3.2.

Statistical tests to calculate *p* values are always indicated in the figure legends. Šidák's multiple comparisons test was always two-sided and used α = 0.05. Tukey's multiple comparisons test and Games-Howell's multiple comparisons test were always calculated at α = 0.05.

**Reporting summary**. Further information on research design is available in the Nature Research Reporting Summary linked to this article.

## Data availability
Raw sequence data from clinical specimens has been deposited in the European Genome-Phenome Archive (accession EGAS00001004646 for 3 PTCL-TFH cases, accession EGAS00001005015 for 15 PTCL-NOS cases). Access to the raw sequencing data from clinical specimens requires the completion of an application form, which will be reviewed by the appointed Data Access Committee, and requestors have to comply with the Data Access Policy. Raw sequence data from murine CD4+ T cells is available in the European Nucleotide Archive (accession PRJEB42764). Sequence data from anaplastic large cell lymphoma was obtained from the Sequence Read Archive (accession SRP044708) and sequence data from healthy lymph nodes was obtained from ArrayExpress (accession E-MTAB-2836). Expression data for immune cells in healthy individuals used in Supplementary Fig. 2b were obtained from DICE. The source data underlying Figs. 2, 3, 4, 5, 6, 7, 8, 9, 10 and Supplementary Figs. 1–6 are provided as a Source Data file. All the other data supporting the findings of this study are available within the article and its supplementary information files and from the corresponding author upon reasonable request. Source data are provided with this paper.

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

## Acknowledgements

This work was supported by Fonds Tom Debackere voor Lymfoomonderzoek and Stichting Tegen Kanker (2018/1272). The Leica SP8x confocal microscope was provided by InfraMouse (KU Leuven-VIB) through a Hercules type 3 project (ZW09-03). K.D. holds a PhD fellowship aspirant of Fonds Wetenschappelijk Onderzoek Vlaanderen (FWO, 1147319N) and an Emmanuel van der Schueren fellowship from Kom Op Tegen Kanker. S.D. holds a post-doctoral fellowship from Stichting Tegen Kanker.

## Author contributions

K.D. conceived the project, collected clinical samples, performed bioinformatic analysis, planned and performed experiments, interpreted data and wrote the manuscript; L.Ma., M.v.B., N.M., O.G., K.J. and M.B. performed experiments; S.D. performed bioinformatic analysis; G.G., L.Mi., C.G. and I.W. provided clinical samples; P.G., L.d.L. and T.T. provided clinical samples and reviewed histopathology; J.C. and D.D. supervised the project, planned experiments, interpreted data and wrote the manuscript.

## Competing interests

D.D. holds a chair funded by Roche. All other authors declare no competing interests.
