## [Peer Review File · Nature Communications]

REVIEWERS' COMMENTS

Reviewer #1 (Remarks to the Author):

The authors have carefully addressed all my comments and have sufficiently answered all question in the revised version. Specifically, they added convincing mechanistic insights concerning the FYN-TRAF3IP2 fusion. They also added important new in vivo data for KHDRBS1-LCK fusion, even though the relevance is not quite as clear, since it was only found in 1 of 55 cases (in total) and also the disease penetrance in mice is low. Nevertheless, the study identifies and characterizes important new lesion that promote the development of peripheral T cell lymphoma (PTCL).

Two minor points:

Supp. Figure 3c upper panel should contain Jurkat instead of Ba/F3 cells.

Figure 7c is not mentioned and described in the text.

Reviewer #4 (Remarks to the Author):

In this work, Debackere et. al. describe two novel, oncogenic fusions, observed in approximately 10% of PTCL, NOS, that "hijack" T-cell receptor (TCR) signaling. While the work presented is novel, it is also consistent with a growing body of evidence implicating the TCR in T-cell lymphomagenesis. Overall, the work presented is rigorous and represents an important contribution to our understanding of PTCL, NOS. While deficiencies were noted in previous reviews of this manuscript, in this reviewer's opinion, these have been adequately addressed in this resubmission.

Minor suggestion: Figure 5c is difficult to see (e.g. visualize histiocytes/eosinophils).

REVIEWERS' COMMENTS

Reviewer #1 (Remarks to the Author):

The authors have carefully addressed all my comments and have sufficiently answered all question in the revised version. Specifically, they added convincing mechanistic insights concerning the FYN-TRAF3IP2 fusion. They also added important new in vivo data for KHDRBS1-LCK fusion, even though the relevance is not quite as clear, since it was only found in 1 of 55 cases (in total) and also the disease penetrance in mice is low. Nevertheless, the study identifies and characterizes important new lesion that promote the development of peripheral T cell lymphoma (PTCL).

We thank the reviewer for the positive evaluation and the interest in this study.

Two minor points:

Supp. Figure 3c upper panel should contain Jurkat instead of Ba/F3 cells.

We thank the reviewer for bringing this to our attention and have corrected supplementary figure 3c accordingly.

Figure 7c is not mentioned and described in the text.

We now refer to figure 7c in the main text.

Reviewer #4 (Remarks to the Author):

In this work, Debackere et. al. describe two novel, oncogenic fusions, observed in approximately 10% of PTCL, NOS, that "hijack" T-cell receptor (TCR) signaling. While the work presented is novel, it is also consistent with a growing body of evidence implicating the TCR in T-cell lymphomagenesis. Overall, the work presented is rigorous and represents an important contribution to our understanding of PTCL, NOS. While deficiencies were noted in previous reviews of this manuscript, in this reviewer's opinion, these have been adequately addressed in this resubmission.

We thank the reviewer for the positive evaluation and the interest in this study.

Minor suggestion: Figure 5c is difficult to see (e.g. visualize histiocytes/eosinophils).

We have now increased the resolution of the embedded image.